# Multi-Omic Identification of Venom Proteins Collected from Artificial Hosts of a Parasitoid Wasp

**DOI:** 10.3390/toxins15060377

**Published:** 2023-06-03

**Authors:** Kaili Yu, Jin Chen, Xue Bai, Shijiao Xiong, Xinhai Ye, Yi Yang, Hongwei Yao, Fang Wang, Qi Fang, Qisheng Song, Gongyin Ye

**Affiliations:** 1State Key Laboratory of Rice Biology and Breeding, Ministry of Agricultural and Rural Affairs Key Laboratory of Molecular Biology of Crop Pathogens and Insects, Key Laboratory of Biology of Crop Pathogens and Insects of Zhejiang Province, Institute of Insect Sciences, Zhejiang University, Hangzhou 310058, China; yukaili17@zju.edu.cn (K.Y.); chenjinss@zju.edu.cn (J.C.); xuebai@zju.edu.cn (X.B.); xiongshijiao@zju.edu.cn (S.X.); yexinhai@zju.edu.cn (X.Y.); yylqy@zju.edu.cn (Y.Y.); hwyao@zju.edu.cn (H.Y.); wangf121@zju.edu.cn (F.W.); fangqi@zju.edu.cn (Q.F.); 2Division of Plant Science and Technology, College of Agriculture, Food and Natural Resources, University of Missouri, Columbia, MO 65211, USA; songq@missouri.edu

**Keywords:** *Habrobracon hebetor* venom protein, proteomics, transcriptome, artificial host

## Abstract

*Habrobracon hebetor* is a parasitoid wasp capable of infesting many lepidopteran larvae. It uses venom proteins to immobilize host larvae and prevent host larval development, thus playing an important role in the biocontrol of lepidopteran pests. To identify and characterize its venom proteins, we developed a novel venom collection method using an artificial host (ACV), i.e., encapsulated amino acid solution in paraffin membrane, allowing parasitoid wasps to inject venom. We performed protein full mass spectrometry analysis of putative venom proteins collected from ACV and venom reservoirs (VRs) (control). To verify the accuracy of proteomic data, we also collected venom glands (VGs), Dufour’s glands (DGs) and ovaries (OVs), and performed transcriptome analysis. In this paper, we identified 204 proteins in ACV via proteomic analysis; compared ACV putative venom proteins with those identified in VG, VR, and DG via proteome and transcriptome approaches; and verified a set of them using quantitative real-time polymerase chain reaction. Finally, 201 ACV proteins were identified as potential venom proteins. In addition, we screened 152 and 148 putative venom proteins identified in the VG transcriptome and the VR proteome against those in ACV, and found only 26 and 25 putative venom proteins, respectively, were overlapped with those in ACV. Altogether, our data suggest proteome analysis of ACV in combination with proteome–transcriptome analysis of other organs/tissues will provide the most comprehensive identification of true venom proteins in parasitoid wasps.

## 1. Introduction

Hymenoptera constitutes a mega-diverse insect order with 23 extant superfamilies, many of which are parasitoid wasps [1,2,3]. Many species of parasitoid wasps are currently reported to provide biological control services in agroecosystems [4,5]. Parasitoid wasps are one of the largest groups of venomous animals, in which venoms have evolved independently in as many as two dozen lineages to serve predation, defense, communication, and competition [1,3,6,7,8]. Parasitoid wasp venom, as one of key female-associated virulence factors, is stored in a sac-like reservoir and injected into the host hemocoel during parasitism, in order to create a favorable environment for the development of parasitoid progeny [9]. Venoms commonly consist of a complex mixture of peptides, proteins, and other non-proteinaceous compounds [10,11]. The role, composition, and action modes of venom proteins depend largely on the parasitic life strategy [2,12]. Based on their lifestyles, parasitoid wasps are broadly divided into ecto- and endoparasitoids [12]. In general, venoms from ectoparasitoids are often largely involved in inducing a short- or long-term paralysis to immobilize hosts, block their development following parasitism, and regulate their immunity and metabolism [13,14,15,16,17]. By comparison, endoparasitoid venoms rarely cause paralysis but trigger a highly varied set of alterations, interfering with the host immune system and development or synergizing the effects of other maternal factors (e.g., polydnaviruses) introduced into the hosts [9,10,12,18]. Beyond understanding the mechanisms of host/parasitoid relationships, research into parasitoid venoms has the potential to uncover a wealth of biomolecules in agriculture and pharmacology [10,19]. Therefore, it is necessary and meaningful to accurately reveal the composition and biological functions of parasitoid venoms.

To date, only a few proteins have been individually identified and characterized from the venoms of a restricted number of parasitoid wasp species through traditional methods such as column chromatography separation, gene cloning, and expression methods [10,20]. More recently, the advent of high-throughput technologies of RNA and proteins has greatly contributed to research in identifying and gaining a better understanding of the diversity of venom proteins/peptides from various animals through a “multi-omic” approach, often denoted as venomics, which is the integration of genomics, transcriptomics, and proteomics [6,21,22,23]. For parasitoid wasps, such an integrated approach combined with transcriptomic and proteomic analyses was first utilized to identify and characterize *Chelonus inanitus* venom proteins, which discriminated proteins between the venom gland cells and the venom [24]. Subsequently, this approach has broadly led to insights into identification and characterization of venom proteins from various parasitoids, such as *Hyposoter didymator* [25], *Aphidius ervi* [26], *Pteromalus puparum* [27], *Toxoneuron nigriceps* [2], *Cotesia chilonis* [19], *Pimpla turionellae* [3], *Pachycrepoideus vindemmiae* [28], and *Torymus sinensis* [29]. Nevertheless, although they all use the transcriptome–proteome analyses, there are slight differences in the identification of screening criteria between individuals, with different fold changes of differentially expressed genes (DEGs) and some differing e-values. In addition, several studies have also previously investigated the complex components of venoms from parasitoid wasps by the separate use of proteomic or transcriptomic approaches combined with bioinformatic analyses [30,31]. For example, it was proposed that, for a contig to be qualified as a venom gene, it must be among the 500-top-expressed contigs in the venom transcriptome if its encoding protein contains a predicted signal peptide, or among the top-100-expressed contigs if its corresponding protein lacks a signal peptide [31]. However, venom proteins, identified by the methods mentioned above, may not be entirely genuine components injected into the host during parasitism, for example, ribosomal proteins and eukaryotic translation initiation factor, for venom gland cellular function, metabolism, and physiology [32]. Therefore, it is better to collect the venom fluid directly flowing from the ovipositor of the parasitoid wasp in the egg-laying process inside or outside the host. For honeybee and several social wasps, the venom has been successfully collected through the use of electrical stimulation (ESV) by a device with a mild electric shock [33,34]. In contrast, this ESV method has never been attempted for parasitoid wasps because of their tiny bodies. Limited drops of venom were collected from *Habrobracon hebetor* (synonym: *Bracon hebetor*) by mechanically stimulating the wasp abdomen [35], but mechanical stimulation could not provide sufficient quantities of venom drops to meet the need for identification and characterization of venom proteins.

*H. hebetor* is an ectoparasitoid with a worldwide distribution, a wide host range, and a rapid life cycle [36]. It provides excellent biological control services, both in the agro-ecosystem and post-harvest system [36,37]. The quest to individually identify and characterize its venom components can be traced to the 1970s, of which most attempts have been frustrated. Regardless, the potency of its venom has been recognized for many years [38,39]. In recent years, the transcriptome of female venom glands (VGs) of this wasp was analyzed, and the full-length open reading frames (ORFs) of calreticulin, venom acid phosphatase Acph-1-like protein, and arginine kinase proteins were identified [39]. Similarly, the venom gland transcriptome of this wasp was analyzed bioinformatically, revealing a number of bioactive genes, such as those encoding venom acid phosphatase, trypsin-2, and C-terminal domain nuclear envelope phosphatase, which were related to host immunosuppressive activities [40]. Nevertheless, the full picture of protein components in venom naturally injected into the host still remains to be revealed. In the current study, we directly collected the venom of *H. hebetor* from artificial hosts by mimicking the process of oviposition and venomous injection by the wasp so as to gain the real venom. Through proteomic analysis, we identified venom proteins from artificially collected venom (ACV) and compared them with those in venom reservoirs (VRs) (control), on the basis of our high-quality de novo genome assembly and gene annotation for this wasp [41]. To verify the accuracy of proteomic data, we also collected VGs, Dufour’s glands (DGs), and ovaries (OVs), and performed transcriptome analyses. For venom protein identification efficiency, different methods used for the venom protein identification were compared. Our study provides new insights into an integrated profile of *H. hebetor* venom proteins, and proposes an approach to selecting the criterion for screening venom proteins or genes with multi-omic analyses to enhance both integrity and accuracy of venom protein identification in parasitoid wasps.

## 2. Results

### 2.1. Identification and Analysis of Artificially Collected Venom Proteins Using Method 1

The wasps parasitized a natural host larva (Figure 1A) and an artificial host (Figure 1B). In both images of Figure 1A,B, the outstretched ovipositor pins of the parasitoid wasp can be clearly seen. This indicated that our constructed artificial host also attracted the simulated parasitism by *H. hebetor* and its venom could be collected through the artificial host. The morphology of the VG and VR of the *H. hebetor* is illustrated in Figure 1A. The venom gland consists of eight primary glandular ducts surrounding the VR, and possesses thick muscle tissues and well-developed helical ridges. VRs of parasitoid wasps that can paralyze their hosts have a thick chitin lining, perhaps because their venom components affect neuronal or neuromuscular transmission, potentially also impairing their own neuromuscular function [42].

For proteomic identification, ACV obtained from the artificial hosts was separated by sodium dodecyl sulfate polyacrylamide gel electrophoresis (SDS-PAGE). Six conspicuous different bands and several inapparent bands were observed, with molecular masses ranging from 14.4 to 116 kDa (Figure 2A). The most abundant bands were at 66.2 kDa, 25 kDa, and 14.4 kDa. All the proteins were digested by trypsin and identified using LC-MS/MS via the protein database from the *H. hebetor* genome [41] using Sequest [43]. Similarly, venom proteins obtained from VR were also separated by SDS-PAGE, revealing multiple distinct bands with molecular masses ranging from less than 14.4 kDa to greater than 116 kDa, while the most abundant bands were slightly above 66.2 kDa (Figure 2B), and then subjected to proteome analysis.

A total of 204 single putative proteins were identified from the ACV proteome, of which 200, 176, and 157 putative protein encoding genes were also expressed in VG, DG, and OV transcriptomes, respectively (Figure 3A). Functional classification of the 204 ACV proteins identified 69 enzymes (EN) (33.82%), 11 protease inhibitors (PI) (5.39%), 8 recognition and binding proteins (RBP) (3.92%), 99 other proteins (OTS) (48.53%), and 17 unknown proteins (UN) (8.33%) (Appendix A, Figure 3B). The proportion of other proteins was the largest, and included immune-related proteins, calmodulin, structure-related protein, actins, diapause-related proteins, hexamerin-like protein, storage-related proteins, and arylphorin (Appendix A). Next were enzymes, among which hydrolases were the most abundant, including peptidase, phospholipases, and glycosidases (Figure 3C). Additionally, the protease inhibitors included ten serpin-type inhibitors and one trypsin inhibitor (Figure 3D).

Among 204 proteins identified in the ACV proteome, one unknown protein, Hheb017020.1, was found only in ACV, whereas its encoding gene *UP1702* was not detected in the transcriptomes of VG, DG, and OV (Figure 3A). However, quantitative real-time polymerase chain reaction (qPCR) results showed that it was highly expressed in VG as compared with DG or OV and carcasses (CA) (Figure 3E). Additionally, three other proteins, identified in the ACV proteome by BLASTP as histone H2B630, H2B680, and H2B740, were all identified in the VR proteome, but their corresponding genes were not detected in the VG transcriptome (Figure 3A). The unigene *H2B630* was found to be highly expressed in the OV transcriptome compared to the transcriptomes of the other three tissues (Figure 4Aa), which followed the same trend as the qPCR results. The qPCR results for the other two histone proteins showed (Figure 4Ab,c) that they were also highly expressed in OV. This differed significantly from the transcriptome, with *H2B680* being the most highly expressed in DG (Figure 4Ab), and *H2B740* being highly expressed in CA (Figure 4Ac).

DEGs in the VG transcriptome are critical according to the traditional venom candidate gene screen [19,27,28,30]. We used three replicates from VG samples and three replicates from CA samples to form six cDNA libraries, which were then sequenced. After the raw reads were filtered to eliminate low quality reads, we obtained 56,956,058 bp, 63,124,164 bp, 59,111,910 bp, 60,026,484 bp, 67,645,280 bp, and 66,679,252 bp sequenced clean reads from VG1, VG2, VG3, CA1, CA2, and CA3, respectively. We used Bowtie2 [44] to match the transcriptome reads to the reference genome [41] and obtained 10,992 VG proteins. Based on the BLASTX results, a database for proteomic studies was generated by computationally translating transcriptome sequences into proteins. This was then confirmed through a BLASTP search with the non-redundant protein sequence (NR) at NCBI. DEGs between VG and CA were estimated by DESeq2. The 10,992 proteins in the VG transcriptome were analyzed for differential expression using the screening criteria of |Log_2_ FPKM_VG/FPKM_CA| ≥ 1 and *p*-value < 0.05 (Figure 4B). The differential expression levels of 200 proteins detected in ACV were also labeled (in deepening), allowing the identification of both highly expressed proteins and less-expressed proteins in VG, as well as proteins that were not significantly different (Figure 4B).

### 2.2. Identification of Putative Venom Proteins Using Method 2

As described in Materials and Methods for Method 2, putative venom proteins that are secreted into VR and expressed within VG were identified using the criteria described previously [19,27,28,30,45]. To control false positive rates, expression level cutoffs were set to FPKM_VG > 10, VG to CA expression ratio (FPKM_VG/FPKM_CA) ≥ 2, and corrected *p*-values < 0.001 to define differentially expressed proteins in VG. Based on this criterion, 1803 proteins were defined to be differentially expressed in VG relative to CA (Figure 5A). A total of 1957 putative venom proteins were screened using a combined analysis of the VG transcriptome and VR proteome, and 1289 unigenes were predicted with signal peptide according to SignalP analysis (Figure 5A). To summarize, a total of 152 putative venom proteins were identified as secreted proteins in the VR proteome, and were highly and differentially expressed in VG (Figure 5A). These 152 putative venom proteins were categorized into 73 EN (48.02%, including 61 hydrolases, 1 isomerase, 1 transferase, and 10 oxidoreductases), 3 PI (1.97%), 9 RBP (5.92%), 41 OTS (26.97%), and 26 UN (17.11%). These 152 proteins are detailed in Appendix A, of which hydrolases accounted for 43% of the total protein, and phosphatase, serine protease, and metalloproteinase were the most abundant. Among them, three serine proteases, four metalloproteinases with very high expression, three phospholipase A2 (PLA2) with very high expression, and eight acid phosphatases were found. For acid phosphatases, our results were different from the only one that was identified using the VG transcriptomic analysis by Hussain et al. (2019) in the same wasp (Appendix A).

Nevertheless, this screening method was not able to cover all proteins of ACV, in which some were not highly expressed in VG and had no signal peptide (Figure 4B and Figure 5B). As seen in Figure 5B, 53 ACV proteins were detected to be highly differentially expressed in the VG transcriptome, 120 in the VR proteome, and 97 with signal peptides within the black wireframe. However, only 30 ACV proteins meeting the key criteria of this method were not only highly differentially expressed but also had signal peptides within the white line box (Figure 5B). In contrast, 35 of the remaining ACV proteins were not included in this screening, which did not meet any of these criteria (Figure 5B). After that, 200 out of 204 ACV proteins expressed in the VG transcriptome were further screened by adjusting the differential expression fold (Figure 5C–F). It was found that more than half of the intersecting proteins between differentially expression proteins in VG and ACV contain signal peptides. The number of intersecting proteins increased with the decrease in the differential expression fold regardless of the presence or absence of signal peptides. Even when the differential expression fold was equal to or more than 2, only 54 of 204 ACV proteins, accounting for 26.47%, were covered by the proteins differentially expressed in VG (Figure 5F). Thus, it seems clear that the proteins identified with the criteria of protein differential expression in VG vs. CA combined with the presence of signal peptides could not reasonably reflect all venom proteins that were actually injected into the host.

### 2.3. Identification of Putative Venom Proteins Using Method 3

A total of 148 putative venom proteins, marked in red, were identified (Figure 6A) using Method 3. All 148 proteins were identified in the VR proteome, in which the expression levels of 111 proteins with signal peptides in the VG ranked in the top 500. In contrast, the other 37 without signal peptides ranked in the top 100 (Figure 6A). Details of these 148 proteins are shown in Appendix A. Then, the proteins with or without signal peptides encoded by genes expressed in the VG with different expression levels were extracted for cross-tabulation with 200 of the 204 ACV proteins, of which transcripts were detected in the VG transcriptome (Figure 6B–F). The number of ACV proteins covered by proteins expressed in VG increased with the number of genes expressed in VG from the top 50 to the top 1000. Furthermore, the number of shared proteins containing signal peptides was greater than that without signal peptides, as the genes ranked in the top 500 or above in terms of expression levels (Figure 6B–E). In contrast, only 59 of the 200 ACV proteins, accounting for 28.92%, were among the top-1000-expressed genes in VG transcriptome (TEC_1000), in which the proteins with signal peptides were fewer than those without signal peptides (Figure 6F). Thus, it also seems that the criterion using the expression levels of genes in VG, with the presence or absence of signal peptides, is not scientifically sound for screening and defining the putative venom proteins, although they were actually injected into the host.

### 2.4. Comparison of Venom Putative Proteins Identified Using Different Methods

With the methods mentioned above, more than half of ACV proteins were not screened and identified in Methods 2 and 3 (Figure 3A, Figure 5 and Figure 6). Using Method 2 for integrated analyses with the VG transcriptome, VR proteome, and signal peptide prediction data, only 26 putative venom proteins overlapped in the ACV proteome (Figure 5B and Figure 7A, Appendix A). Among these proteins, there were six hydrolases, three oxidoreductases, two PI, four RBP, one UN, and ten OTS (Appendix A). Similarly, only 25 putative venom proteins (Figure 7A) were shared with those in the ACV proteome analyzed using Martinson’s screening criteria, as described in Method 3 (Figure 7A). Among these proteins, there were four hydrolases, one oxidoreductase, two PI, five RBP, and 13 OTS (Appendix A). Excluding overlapping proteins between Method 2 and Method 3, there were 169 ACV proteins that could not be identified in Method 2 and 3 (Figure 7A). Thus, it is clear that these two methods could not fully uncover the composition of the putative venom proteins that were actually injected into the host.

In contrast, a total of 200 out of 204 ACV proteins were expressed in the VG transcriptome when the signal peptide and expression analysis of venom proteins were not considered (Figure 7B). In addition, the transcripts of four ACV proteins, including three histones and one unknown protein, were not found in the VG transcriptome, but they were more or less expressed in VG as detected using qPCR, of which one unknown protein was highly expressed in VG (Figure 3E), but three histones did exist in OV when compared with other tissues (Figure 4A). It seems as if the expression levels of these four proteins were too low to be detected in the VG transcriptome. As 120 putative venom proteins existed in the VR proteome, they only covered 58.82% of all ACV proteins (Figure 7B). The possible reason for this is that the proteins in VR are full of released venom proteins, but not all venom proteins from VG were fully released into VR in the absence of parasitic processes. Another reason is that the VR proteins with higher abundance may cause those with lower content to be undetectable during the proteomic detection process. Thus, it is clear that the method with combined analyses of the VG transcriptome and ACV proteome is better for identifying the composition of venom proteins that are actually injected into the host during the parasitism process.

### 2.5. qPCR and PCR for Gene Accuracy

To validate the gene accuracy, the expression levels of 20 selected putative venom proteins in VG were detected by qPCR, and combined with FPKM values from the VG transcriptomic data. The expression of all 20 selected putative venom proteins in VG differed from that in CA (Figure 8A), with broadly similar trends to the transcriptomic analysis results. The expression profiles of 12 putative venom proteins in VG, DG, and OV are shown in Figure 8B–N. Based on the comparison of the qPCR results with the transcriptome FPKM values, the expression patterns were nearly identical for both, demonstrating the reliability of the transcriptomic data. Of these proteins, post-GPI attachment to protein factor 4 (PGAP4) was highly expressed in OV, serine proteinase stubble (SB) was highly expressed in DG, and the rest were highly expressed in VG. All of them were from ACV, except Paralytic protein 1, which was from Venom proteome 1 (Figure 7A and Appendix A), and PGAP4, which was from VR proteome. Six ACV proteins in Figure 8O–T, however, showed significant variations in both relative expression level and FPKM. This could be because the transcriptomes of all four tissues do not contain these six proteins at significant levels, making it difficult for the test to accurately calculate their FPKM values. The majority of proteins were more highly expressed in VG, with the exception of the Cadherin-like protein (Cadherin), according to qPCR data (Figure 8O–T).

In order to check the accuracy of these protein sequences, we also randomly chose 20 proteins without a signal peptide from ACV. These protein-encoding genes were polymerase chain reaction (PCR) amplified using VG cDNA as a template. The amplification bands were displayed in Appendix A and were sent for sequencing. The sequencing results (Appendix A) could be compared to the corresponding genes in the nucleic acid library of VG transcriptome by BLASTN and were proven to be expressed in VG transcriptome.

## 3. Discussion

In this study, we developed a novel venom collection method using artificial host larvae (amino acid encapsuled with membrane, allowing parasitoid wasps to inject venom). The biological activity of ACV obtained through this new venom protein collection method was not affected by the content of the artificial host, and the same was observed for VR protein. The effects of injecting ACV and VR proteome on the host were almost identical to those of natural parasitism (Appendix A), which resulted in immediate paralysis and cessation of development in the host. We identified 204 proteins in ACV via proteomic analysis and genomic protein libraries. The traditional putative venom protein screen was integrated with the VG transcriptome and the VR proteome, and 152 and 148 putative venom proteins were identified through 2 different screening criteria (Method 2 and Method 3), respectively. Compared to ACV, only 16 putative venom proteins overlapped in total (Figure 7A). However, in ACV, 200 proteins were present in the VG transcriptome and 120 were present in the VR proteome (Figure 7B). Four proteins were found to be not included; one of these, UP1702, was highly expressed in the VG versus other tissues, and three histones were highly expressed in the OV in qPCR analysis (Figure 3E and Figure 4A).

There were significant differences in the putative venom proteins identified between ACV and traditional methods, and we analyzed the advantages and disadvantages of each of these three different methods for identifying putative venom proteins. First, not all venom proteins included signal peptides. From the previous PCR validation, genes encoding proteins without signal peptides were present in VG, indicating that these proteins would be expressed in VG. Our results showed ACV is the best source for identifying putative venom proteins that are truly injected into the host, compared to other sources. It was also found that the 33 kDa venom protein in *C.* sp. near *curvimaculatus* has a positively charged N-terminal, which was in contrast to the classical signal peptides containing basic residues, and it is speculated that Hymenoptera may have evolved specialized protein processing and secretion pathways [20,46]. A researcher reported the involvement of a channel system linked via actin in the VG of five parasitic wasps in the synthesis and release of venom parasitic factors [47]. Therefore, the proteins without a signal peptide could also be secreted into the host by the parasitoid wasp. Secondly, ion suppression could seriously interfere with the ionization and detection of analytes in proteomics [48]. Due to the abundance of proteins in VR, low levels of protein may not be detected in the VR proteome. Following the traditional approach [19,27,28,30,45], we filtered out proteins that were expressed at low levels in VG. It is possible that not all venom proteins required extremely high expression levels. In addition, some proteins might play a role at low expression levels to assist parasitic wasps in their parasitism, or be expressed in the whole insect and simply perform different functions in different tissues. Alternatively, the protein may not be highly expressed at the time point sampled, but only after conditions have been met. Venom composition and content were influenced by the age of the parasitoid wasps and the incidence of parasitism [49]. Similarly, post-translational modifications (PTMs) could not be detected by transcriptomics [48]. Furthermore, in Hymenoptera the venom apparatus was associated with the female ovipositor, and its ancestral function was probably associated with the coating of eggs [11]. We speculated that not all the proteins we collected by mimicking parasitized hosts were necessarily secreted from VG, but might also came from the female reproductive system, such as DG, OV, oviducts, and accessory glands, just as three histones might come from OV. Therefore, the proteomic methods used in this article might have missed small peptides and fragments of proteins, as well as proteins with low expression, and the parasite factor proteins identified by these methods might be incomplete. Moreover, because most hymenopteran parasitoids are very small insects that produce minute quantities of venom [19], it might not be convenient to collect venom proteins from VR and artificial hosts. Two conventional venom protein screening methods are both related to venom gland transcriptome analysis, so it might be possible to obtain parasite-associated factor proteins using the venom gland transcriptome. Therefore, we proposed that the putative venom protein can be identified directly using the VG transcriptome, which can disregard signal peptides and expression levels. However, this method has a limitation, which is that the number of obtained proteins was too high, including potential interference from tissue cells, hindering the accurate screening of putative venom proteins. In addition, we found some proteins in the ACV that were similar to cell contents, and it cannot currently be determined whether they are venom proteins. From the perspective of venom acquisition methods, the influence of tissue fragments in ACV was minimal compared to VR proteome. However, if these proteins are not secreted as venom proteins, does this indicate that *H. hebetor* venom also has a holocrine secretion mechanism, like *Phoneutria nigriventer* [50]? Or does it have venom vesicles, similar to *Drosophila* parasitoid wasps [51]? These cellular structures might be secreted into the artificial host with the venom, thus affecting our identification of putative venom proteins in ACV. Further experiments are required to verify hypotheses.

There were 16 putative venom proteins in ACV that overlapped with those identified through the combination of VG transcriptomic and VR proteomic screening methods (the two different screening criteria are respectively referred to as Methods 2 and 3 in this article). These proteins included one oxidoreductase called superoxide dismutase; three hydrolases including phospholipase A2-like (PLA2), venom carboxylesterase-6, and neutral alpha-glucosidase AB; two PI including alaserpin isoform X5 and leukocyte elastase inhibitor isoform X1; three RBP including two DN1895 and one DN4395; and seven OTS including tetrapeptide repeat homeobox-like (TPRXL) isoform X2, kielin/chordin-like protein, calreticulin, venom allergen 3-like, graves disease carrier protein-like, protein quiver-like, and endoplasmic reticulum chaperone BiP (Appendix A). In the case of PLA2, for example, we identified a total of four PLA2 proteins as possible venom proteins in the ACV using Methods 2 and 3 (Appendix A). One of them, PLA2_2, was common to ACV (Methods 2 and 3), and highly expressed in VG with a signal peptide (Appendix A). Another PLA2_1 with a signal peptide was common to Methods 2 and 3, and again highly expressed in VG (Appendix A). PLA2_4 was from ACV with a signal peptide, but was not expressed at high levels in VG (Appendix A). In contrast, PLA2_3 from Method 2 had a signal peptide and was differentially highly expressed in VG (Appendix A). Two of the four subunits of the venom paralytic toxin BrhTX-1 [52] were PLA2s by BLASTP; however, only BrhTX-1(c) of the two subunits were PLA2_3, which was the venom protein identified using Method 2 (Appendix A). BrhTX-1(b) was not identified as a venom protein (Appendix A). The possible reasons for this are that it has no signal peptide and the protein has a molecular weight of approximately 21.04 kDa, which may had been filtered out during the collection of ACV. Additionally, four PLA2s from *H. hebetor* were compared in multiple sequences with those of *A. mellifera*, *C. insularis*, *Fopius arisanus*, and *Diachasma alloeum* (Appendix A), and compared to the PLA2s from honey bees; the PLA2s in these parasitoids in the C segment lacked two conserved cysteine residues [53], but both had two conserved structural domains.

Although there was little overlap between ACV and Methods 2 and 3, our analyses of the composition of all the putative venom proteins revealed that some shared the same annotated names. ACV had 26 and 25 identical IDs to the putative venom proteins identified in Methods 2 and 3, respectively (Figure 7A). In addition, there were 100 identical IDs in Methods 2 and 3 (Figure 7A). The combined analysis revealed that a total of 369 different IDs of proteins were identified (Appendix A). These proteins were from 162 protein families (Appendix A). There were eighteen protein families in total across the three methods, with serpin and trypsin being the most numerous. This was closely related to the function of the venom protein, which suppresses host immunity and regulates host development and metabolism [12]. There are 73 protein families unique to ACV proteins, with 11 histone proteins, which comprised their largest family (Appendix A). The identification of histone H2B 3-like in the honey bee venom collected by electrical stimulation has been reported [33], but unfortunately its function has not been investigated. In the recent studies of parasitoid wasp venom proteins, the histone protein was reported as a venom protein only in *Chouioia cunea* [54], and the second most abundant proteins were actin proteins. Actin might be involved in the transport of venom proteins in parasitoid wasps and in their hosts [47].

In addition, the putative venom proteins identified by ACV and Methods 2 and 3 in this paper were analyzed against the 61 *H. hebetor* toxins and proteins from VG tissues, and all of them had been reported to be uploaded to the National Center for Biotechnology Information (NCBI) (Appendix A) [38,52,55,56]. Twenty of these proteins had signal peptides, and, of course, three of the short peptides reported by Quistad et al. (1994) were not complete [38]. However, we found these three complete proteins with signal peptides based on our VG transcriptome protein library by BLASTP (Appendix A). In this study, we identified Brh-III as a venom protein using Method 2, while Brh-I and Brh-V were not identified as venom proteins. Unfortunately, Brh-III (Paralytic protein 1) was not found in ACV, probably because it was too small and was probably filtered out during the collection of ACV. The other two short peptides, Brh-I and Brh-V, which were also not detected in ACV, possibly for the same reasons, were derived from two different arylphorin subunit alphas that might have undergone shear modification to become venom short peptides. Four proteins were unable to be found in the VG protein library, three of which were reported by Zurovec et al. (2017) to be present in VG, and one toxin that was from Windass et al. (1996) [52] (Appendix A). Even within the same parasitoid wasp species, there were differences in the venom protein fractions of the different populations [57]. In addition, N-terminal sequencing may not be accurate. A total of 18 proteins were detected from ACV and Methods 2 and 3 in this paper (Appendix A). Four of these proteins were common to ACV and Method 2 and 3 (Appendix A). In 2016, the VG transcriptome of *H. hebetor* was reported and a full-length ORF of calreticulin was identified [39]. In this study, the same calreticulin was also found using ACV and Methods 2 and 3 (Appendix A and Figure 6A). In addition, five proteins belonged to ACV, including two protease inhibitors, two hydrolytic enzymes, and one actin (Appendix A). Five proteins belonged to Method 2 but three of these proteins were found to be Paralytic protein 1 after BLATP (Appendix A). Two proteins belonged to Method 3 and two proteins were common to Methods 2 and 3 (Appendix A). In 2019, Hussain et al. isolated and identified two venom acid phosphatases [40], one of which was different from the venom acid phosphatase protein identified in this paper, and the other was identified as a venom protein using Methods 2 and 3 (Appendix A). The protein uploaded by Zurovec et al. (2017) [56] also had two venom acid phosphatases; unfortunately, we did not identify these as venom proteins, but both were present in the VG transcriptome (Appendix A).

Additionally, we also compared the putative venom proteins with other parasitic wasps. In previous studies, the venom gland transcriptome of *H. hebetor* and *Bracon nigricans* showed only a few common transcripts [58]. In the present study, the following proteins including venom acid phosphatase, and PLA2 and trypsin-like serine protease were also identified in *H. hebetor* (Appendix A), which were similar to those in *B. nigricans*. Arginine kinase Hheb077500.1 was found in *H. hebetor* (Appendix A) and *B. nigricans*, but we did not classify it as a venom protein in this paper using the three methods used. This might be due to the absence of signal peptides and the low expression level in VG. Serine protease inhibitors control functions such as the immune system, digestion, and protection against predators in insects [59]. In ACV, four serpin proteins were identified that were presumed to play an important role in helping *H. hebetor* to suppress the host immune response. Superoxide dismutase (SOD) is an important anti-oxidative stress protein, which in ectoparasitoid *Scleroderma guani* is associated with hemolymphatic melanization in the host [60]. We identified a superoxide dismutase, Hheb020410.1, using different methods (Appendix A, Figure 7A). In addition, we compared the 152 putative venom proteins with the proteins that were obtained by the same method in other species (Appendix A). There was 53.29% orthologous similarity to five other parasitoids: *C. chilonis*, *Microplitis mediator*, *Nasonia vitripennis*, *P. puparum*, and *P. vindemmiae*. There were 35 proteins with orthologous groups in all 6 parasitoid wasps, of which, 28 were hydrolases, such as venom metalloproteinase and transmembrane protease serine 9-like (Appendix A). An overwhelming number of venom components were enzymes with similarities to insect metabolic enzymes; these enzymes are recruited for expression in VG with modified functions [10]. Among the remaining proteins, there were three known venom proteins and four unknown proteins (Appendix A). The functions of these seven proteins have not yet been studied. In comparison to the other three Pteromalidae parasitoid wasps, *H. hebetor* possessed more orthologous groups than the two braconid wasps, with counts of 55 proteins in *C. chilonis* and 58 proteins in *M. mediator* (Appendix A). More putative venom proteins were identified in *H. hebetor* than in other parasitoids. It is possible that *H. hebetor* has more abundant putative venom proteins in order to overcome more host defense mechanisms in co-evolution with hosts. Additionally, both braconid wasps had polydnavirus auxiliary parasites compared to *H. hebetor* [61,62].

## 4. Conclusions

The composition of parasitic wasp venom and the functions and applications of individual venom parasitic factor proteins are promising research topics. Substances that are largely untapped in venom, such as peptides, proteins, and alkaloids, are potentially useful sources of therapy [10,63]. In early studies, it was found that the crude venom of *H. hebetor* had anti-inflammatory and anti-tumor effects, as well as the ability to enhance immunity [64]. Identification of *H. hebetor* venom proteins is the basis for further detailed functional analysis of these venom proteins. ACV proteins (a total of 204) were found to be parasitic factor proteins that were actually injected into the host, except three histones that might come from other tissues; the remaining 201 proteins were putative venom proteins. Further functional studies are needed to determine whether these candidate genes are indeed toxins, and the present research could only identify these proteins as candidate venom proteins.

## 5. Materials and Methods

### 5.1. Insect Breeding and Parasitization

*H. hebetor* was reared on the host larvae of the Indian meal moth, under the laboratory conditions of 27 ± 1 °C and 75% relative humidity (RH) with a 14 h:10 h (light:dark) photoperiod [36]. *P. interpunctella* was raised in the same environment with crushed wheat grains mixed with honey, glycerol, and yeast [65].

### 5.2. RNA Preparation, Complementary DNA (cDNA) Library Construction, and Illumina^®^ Sequencing

*H. hebetor* females within 3–7 days after breaking out of their cocoons were immobilized on ice. Under a microscope, the wasp was infiltrated in phosphate-buffered saline solution (10 mM of 1 × PBS, pH 7.4) and its venom glands (VGs) were removed by grasping the tip of the ovipositor with fine forceps while holding the abdomen with another pair of forceps (Figure 1A). Approximately 200 VGs were isolated individually and transferred to a 2.0 mL RNase-free Eppendorf tube containing 1 mL of Tri-Reagent (catalogue # T9424, Sigma Chemicals) and steel beads (Beyotime Biotechnology, Shanghai, China). After brief shaking to break up the VG tissue, the samples were centrifuged immediately at low speed to ensure that all tissues were submerged in the lysate, and stored at −80 °C. We also prepared three types of transcriptome sample using a similar method, including approximately 200 Dufour’s glands (DGs), 200 ovaries (OVs), and 20 carcasses (CAs, the whole body without VG, DG, and OV) in each sample in a similar way. The shape of the VG, DG, OV, and venom reservoir (VR) of *H. hebetor* are shown in Figure 1A. All these dissected samples were used for RNA-Seq analysis, with three replicates each. The complementary DNA (cDNA) libraries were performed by Nextomics Biosciences (Nextomics, Wuhan, China), followed by sequencing two 150 bp paired-end lanes using an Illumina HiSeqTM system (Illumina, Omaha, NE, USA).

### 5.3. Read Assembly and Unigene Annotation

In a previous study, we sequenced and assembled the *H. hebetor* genome having a size of 131.6 Mb with a contig N50 of 1.63 Mb [41]. For the VG and CA samples (3 replicates each with a total of 6 samples) in the present study, we used the NGSQCToolKit (v2.3.3) for data filtering [66], and FastQC (http://www.bioinformatics.babraham.ac.uk/projects/fastqc/, accessed on 23 July 2018) for quality control. The Q30 were above 93% for all samples, and we then generated ~179.2 million clean reads, for a total of 25,529.32 Mb for VG sequences. Two unfiltered paired-end lanes of each sequence have been deposited as a series in NCBI’s GEO database under the accession number PRJNA971361 or in the NCBI Short Read Archive under submission number. The raw data in fastq format were processed by an internal Perl script to obtain clean data. In this step, reads containing adapters, ploy-N, and inferior quality reads were removed. At the same time, Q20, Q30, GC-content, and sequence repeat levels were calculated for the clean data. The clean data were then assembled with the reference genome [41] using Trinity v2012-10-05 [67]. After assembling, the longest cluster sequences from each transcript were chosen as the reference sequences for subsequent analyses (henceforth, called unigenes). All unigenes were annotated on NCBI nonredundant protein sequences (NR) at the NCBI database using BLASTN and BLASTP with an e-value < 1 × 10^−5^. After gene identification, the signal peptides of putative venom proteins were identified using SignalP v5.0 (http://www.cbs.dtu.dk/services/SignalP/, accessed on 20 October 2021). Both DG and OV transcriptome data were processed in the same way. We generated ~160.4 million clean reads for DG and ~170.0 million clean reads for OV sequences.

### 5.4. Transcriptomic Data Analysis

Gene expression levels in the VG, DG, OV, and CA samples of *H. hebetor* were estimated via RNA-Seq using the Expectation Maximization (RSEM) [68], with expression levels calculated and conveyed as fragments per kilobase of transcript per million reads mapped (FPKM). The expression profiles of putative venom protein genes were visualized using TBtools [69]. Differentially expressed unigenes were defined using DESeq2 (http://www.bioconductor.org/packages/release/bioc/html/DESeq2.html, accessed on 10 September 2021) [70] with strict screening thresholds of a corrected *p*-value < 0.001, log_2_(FPKM_VG/FPKM_CA) > 1 and FPKM > 10 [27].

### 5.5. Protein Collection

Mated *H. hebetor* female wasps aged 3–7 days were anesthetized at −20 °C for 10 min as described above, and then dissected in sterile PBS containing 1 mM ProteinSafe^TM^ Protease Inhibitor Cocktail (Transgen, Beijing, China) on an ice plate under a stereoscope (Leica, Wetzlar, Germany). VR was separated and washed thrice, and then transferred into 1.5 mL Eppendorf tubes. After centrifugation at 16,000× *g* for 20 min, the supernatant was transferred into a new 1.5 mL Eppendorf tube and stored at −80 °C until use. The concentration of the venom proteins was determined by a Modified Bradford Protein Assay Kit (Sangon Biotech, Shanghai, China) according to the manufacturer’s protocol. The same method was used to obtain proteins from DG and OV.

### 5.6. Artificial Host Production and Venom Protein Collection

A 20 μm thick (3-filament) polyethylene film was pressed with a heated glass mill to produce several long protrusions having a diameter of 2–3 mm and height of approximately 20 mm. A mixture of 20 μL of amino acids (Sigma Aldrich, Taufkirchen, Germany) (900 ng/mL leucine, 600 ng/mL phenylalanine, 700 ng/mL histidine) was added to the concave surface, covered with a flat polyethylene film, and heat sealed to create artificial host larvae [71]. The artificial hosts were mixed with the natural host, *P. interpunctella*, for 1 h, allowing them to possess the scent of the natural hosts, and were then inoculated with *H. hebetor* female wasps for 4 h in the dark at the ratio of wasps to hosts of 300:1. After the successful processes of oviposition and venomous injection by female wasps, the artificial hosts were cleaned of surface impurities and the amino acid solution from the artificial hosts was transferred to 3 kDa ultrafiltration tubes by a brief low-speed centrifugation at 500× *g* for 10 s. We used approximately 3000 parasitoid wasps and alternately parasitized artificial hosts for 4 days, 4 h a day, using a total of 40 artificial hosts for venom collection. The amino acid solution was concentrated by centrifugation and replaced with PBS solution. Protein concentrations of collected venom were determined using the bicinchoninic acid (BCA) protein assay kit (Thermo Fisher Scientific, Waltham, MA, USA) and protein profiles were analyzed by 8–16% SDS-PAGE. We collected 16.52 μg of venom protein for mass spectrometry detection.

### 5.7. Analysis of Proteins from VR and ACV Using SDS-PAGE and LC-MS/MS

Proteins from *H. hebetor* VR and ACV were denatured at 100 °C for 15 min and separated by 8–16% SDS-PAGE and stained with Coomassie Brilliant Blue R-250 for quality checks. Two venom proteins in solution were digested with trypsin to form peptides and analyzed on a liquid chromatography tandem mass spectrometry (LCMS/MS) system (LTQ-VELOS, Thermo Finnigan, San Jose, CA, USA). Samples were subsequently desalted on a Zorbax 300 SB-C18 column (Agilent Technologies, Wilmington, DE, USA) and then separated on an RP-C18 column (150 m i. d., 150 mm long) (Column technology Inc., Fremont, CA, USA). Buffer A was water containing 0.1% formic acid, Buffer B was 84% acetonitrile containing 0.1% formic acid. The gradients for Buffer B were: 0–3 min, from 3% to 9%, 3–93 min, from 9% to 32%, 93–108 min, from 32% to 40%, 108–113 min, from 40% to 100%, and 113–120 min. A raw proteome was generated and the Sequest search algorithm [43] was used to search for identified peptide fragments against the protein database set up with the *H. hebetor* genome [41]. Parameters were set as follows: carbamidomethyl was set as a variable modification, and cross-correlation scores (Xcorr) (Charge = 1, XCorr ≥ 1.9, Charge = 2, XCorr ≥ 2.2, Charge = 3, XCorr ≥ 3.75, and delta CN ≥ 0.1) were used as the filter criteria. This part of experiment was conducted by Shanghai Applied Protein Technology Co., Ltd. (Shanghai, China).

### 5.8. Venom Protein Identification

For venom protein identification, the following three methods were employed. Method 1: The putative venom proteins were analyzed and identified with the ACV proteome through screening of the protein database set up with the *H. hebetor* genome [41]. The screening criterion was that putative venom proteins in ACV proteome should be found in the genome-based protein database of this wasp.

Method 2: The putative venom proteins were identified with an integrated analysis using a combined information of the VG transcriptome, VR proteome, DEG levels in VG, and signal peptide prediction. Referencing most previous investigations under the assumption that venom proteins are secreted and expressed within VG [19,27,28,30,45], we identified putative venom proteins using three criteria as follows: proteome supports the predicted protein in VR, expression supports the unigenes in VG, and signal peptide prediction supports the protein with a signal peptide. Differential expression analysis between VG and CA was performed using DESeq2 [70] with strict screening thresholds of a corrected *p*-value < 0.001, log_2_(FPKM_VG/FPKM_CA) > 1, and FPKM_VG > 10 [27]. These DEG-encoding proteins possessing the signal peptides were screened and detected in the VR proteome. Meanwhile, the criteria were adjusted as follows: DEGs were screened with thresholds of a corrected *p*-value < 0.05, log_2_ (FPKM_VG/FPKM_CA) > 1, 2, 4, 8, respectively, and FPKM_VG > 10. The corresponding protein must also be present in the VR, regardless of being with or without a signal peptide.

Method 3: Referencing Martinson et al. (2017) [31], the putative proteins were identified using two major criteria as follows: (1) the corresponding protein must be present in the VR proteome, and its corresponding gene must be expressed in VG; (2) to be qualified as a venom gene, the contig must be one of the top-500-expressed genes in the VG transcriptome, encoding the protein with a predicted signal peptide, or one of the top-100-expressed genes if lacking a signal peptide in its encoding protein. Meanwhile, the criteria were slightly adjusted as follows: combined with the VR proteomic analysis results, the unigenes were qualified as venom genes, which must be one of the top 200, 500, and 1000 expressed genes in the VG transcriptome, respectively, regardless of whether the corresponding protein contains a signal peptide or not.

### 5.9. Compare with the Venom Proteins Screened by Method 2 from Six Parasitoid Wasps

The 152 putative venom proteins of *H. hebetor* identified by Method 2 were analyzed for orthologous groups with the putative venom proteins from the other 5 parasitoids identified by the same method. The putative venom proteins of 5 parasitoid wasps were: 55 in *C. chilonis* [19], 75 in *M. mediator* [45], 79 in *N. vitripennis* [30], 70 in *P. puparum* [27], and 64 in *P. vindemmiae* [28]. OrthoMCL was used to categorize putative venom proteins into orthologous groups, with a cutoff *p*-value of 1 × 10^−5^ [28,72].

### 5.10. Predict the Venom Proteins Families

Venom proteins were predicted in protein families using Pfam 35.0 (http://pfam-legacy.xfam.org/, accessed on 26 December 2022) and HMMER v3.3.2 (http://hmmer.org/, accessed on 26 December 2022), with a cutoff *p*-value of 1 × 10^−3^ and at least one hit.

### 5.11. Multiple Sequence Matching

The protein sequences were compared using the Clustal algorithm (https://www.genome.jp/tools-bin/clustalw, accessed on 5 May 2022), and the sequence comparison results were then plotted using ESPript 3.0 (https://espript.ibcp.fr/ESPript/ESPript/index.php, accessed on 5 May 2022).

### 5.12. qPCR

We performed qPCR on the CFX Connect^TM^ Real Time Detection System (Bio-Rad, Hercules, CA, USA) using SYBRRPremix Ex Taq^TM^ II (Tli RNaseH Plus) (Takara, Japan) and ChamQ SYBR qPCRMaster Mix (Vazyme, Nanjing, China). Total RNA of VG, DG, OV, and CA was separately extracted. cDNA was synthesized from 1 mg total RNA using the TransScript One Step gDNA Removal and cDNA Synthesis SuperMix (Transgen, Beijing, China). All specific primers for qPCR were designed using Primer3 (https://primer3.ut.ee/, accessed on 30 October 2021). The programs were set as follows: enzyme activation at 95 °C for 30 s, followed by 40 cycles, denaturation at 95 °C for 5 s, annealing and extension at 60 °C for 30 s, and a melting curve analysis. mRNA expression levels were normalized to the reference (18S rRNA), and quantified based on the comparative 2^−ΔΔCt^ method [73]. The experiments were repeated 3 times. Statistical analyses were implemented in GraphPad Prism 6 (https://www.graphpad.com/, accessed on 14 March 2023). The primer sequences used in the validation of expression profiling for 43 proteins from the VG transcriptome and ACV are listed in Appendix A.

### 5.13. Sequencing Confirmation

Additionally, using cDNA from VG tissues as a template, we performed polymerase chain reaction (PCR) using KOD One^TM^ PCR master Mix (TOYOBO CO., LTD., Osaka, Japan), gathered PCR products, and carried out next-generation sequencing on 20 ACV protein encoding genes that were highly expressed in the VG transcriptome but lacked a signal peptide. The sequencing results were aligned with the transcriptome assembly sequences. Appendix A provides more information on the primer sequences for these 20 genes.

### 5.14. Crude Venom Injection

Healthy fifth instar *P. interpunctella* larvae were selected. Different concentrations of crude venom were prepared at 1, 0.5, 0.25, and 0.125 VRE, with protein concentrations of 0.83 μg/μL, 0.54 μg/μL, 0.23 μg/μL, and 0.10 μg/μL, respectively. Additionally, venom proteins collected by artificially parasitizing hosts (ACV, 0.12 μg/μL) and 1 μg/μL BSA protein were injected into fifth instar Indian meal moth larvae using a microinjection system (Nanoject II, Drummond scientific company, Broomall, PA, USA), at a volume of 36.8 nL per. Thirty larvae were injected with different concentrations of crude venom. After 30 min of parasitism by *H. hebetor*, the same number of host larvae were transferred to new culture dishes without parasitoid wasps, and the parasitic wasp eggs on the surface of the hosts were removed. Finally, the unparasitized and differently treated hosts were placed in a culture box with temperature of 27 ± 1 °C, 14 h:10 h (light:dark), and relative humidity of 75%, and observed and photographed using a Leica M205 A stereomicroscope (Leica, Wetzler, Germany).

## Figures and Tables

**Figure 1 toxins-15-00377-f001:**
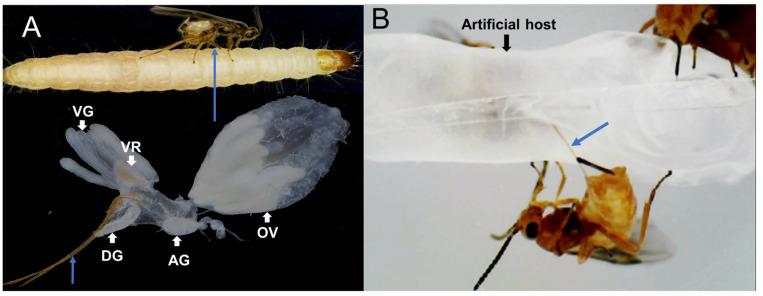
The morphological aspects of the *Habrobracon hebetor* wasp and interaction with a parasitized *Plodia interpunctella* larva and an artificial host. (**A**) *P. interpunctella* larva was parasitized by *H. hebetor*, of which the reproductive organs are shown as ovaries (OVs), and venom glands (VGs) associated with the venom reservoir (VR), Dufour’s gland (DG), and abdominal ganglion (AG). (**B**) The mimicked host was parasitized by *H. hebetor*. The blue arrow points to the wasp ovipositor.

**Figure 2 toxins-15-00377-f002:**
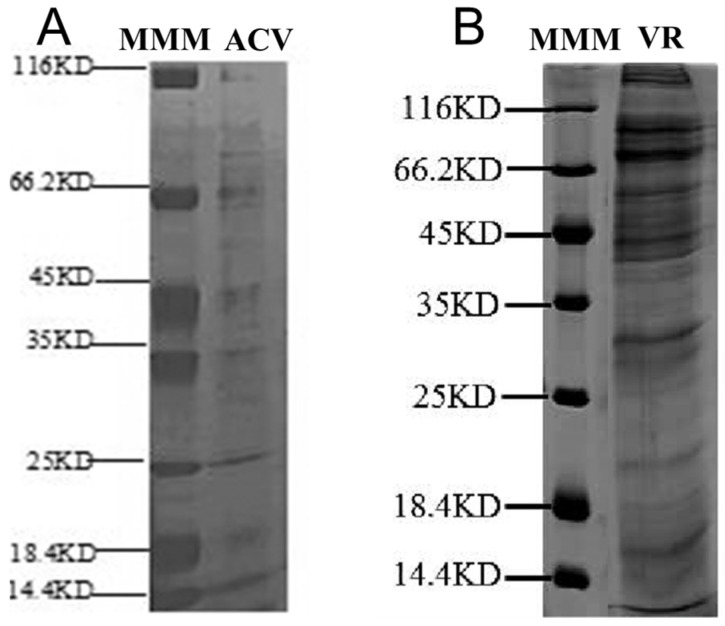
The sodium dodecyl sulfate polyacrylamide gel electrophoresis (SDS-PAGE) analysis of two venom proteins under denaturing conditions. (**A**) Denaturing 8–16% SDS-PAGE analysis of artificially collected venom proteins from *Habrobracon hebetor* followed by Coomassie Brilliant Blue staining. ACV, artificial collection venom from mimicked hosts; MMM, molecular mass markers. (**B**) Denaturing 8–16% SDS-PAGE analysis of *H. hebetor* venom reservoirs (VRs) proteins followed by Coomassie Brilliant Blue staining.

**Figure 3 toxins-15-00377-f003:**
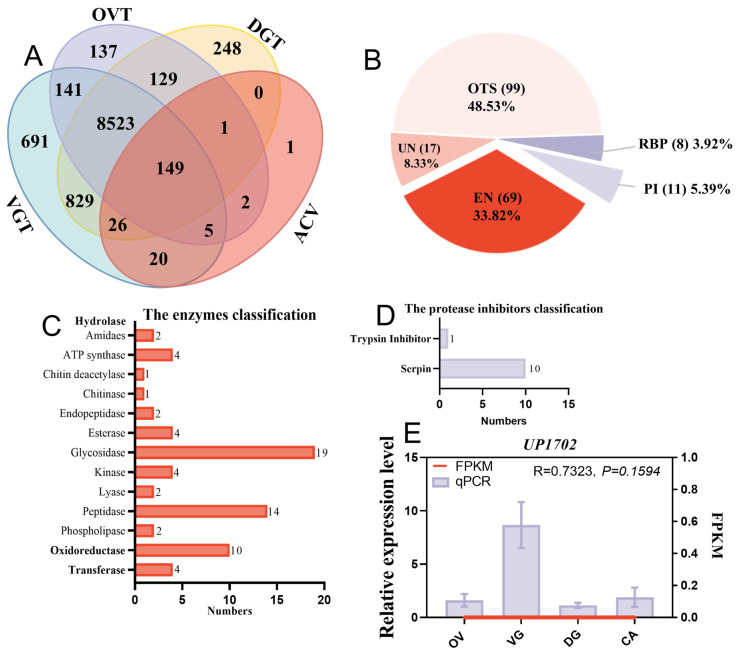
Analyses and identification of artificially collected venom proteins from *Habrobracon hebetor*. (**A**) Venn diagram of putative venom proteins combining transcriptomic and proteomic analyses. VGT, OVT, and DGT, proteins identified using the venom glands, ovaries and Dufour’s glands’ transcriptome, respectively. ACV, artificial collection venom from mimicked hosts. (**B**) Categories and percentage of 204 ACV proteins. EN, enzymes; PI, protease inhibitors; RBP, recognition and binding protein; OTS, other proteins; UN, unknown proteins. (**C**) The classification of the enzymes in ACV. (**D**) The classification of the protease inhibitors in ACV. (**E**) Tissue expression profiling of *UP1702* only found in ACV. *UP1702*, encoding an unknown protein Hheb01702.1; FPKM, fragments per kilobase of transcript per million reads mapped; qPCR, quantitative real-time polymerase chain reaction. The R-values were calculated to analyze the correlation between the FPKM of four tissues’ transcriptomics and the relative expression levels of qPCR, in order to describe their trends of variation in four tissue samples. The *p*-value represented the probability value. OV, ovaries; VG, venom glands; DG, Dufour’s glands; CA, carcasses.

**Figure 4 toxins-15-00377-f004:**
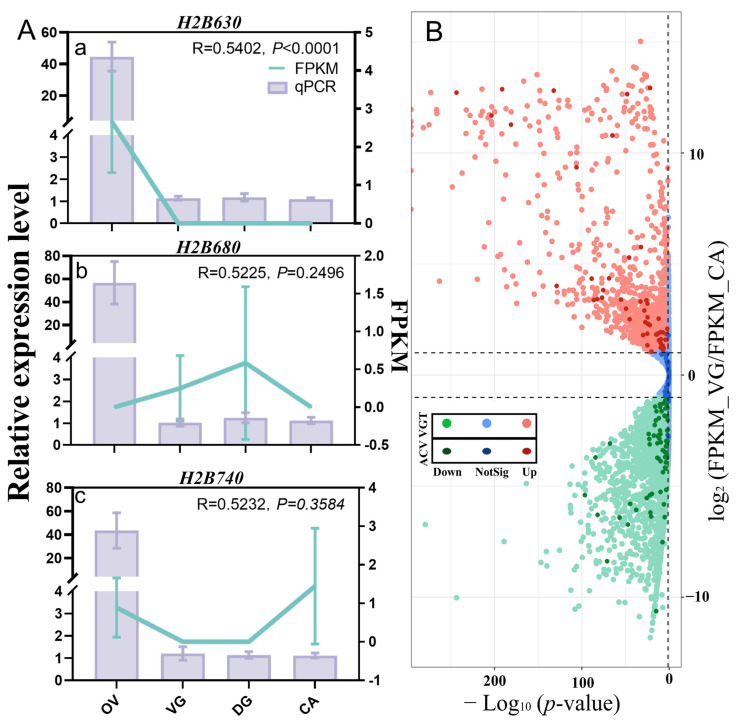
Transcript levels of venom gland proteins measured by qPCR and differential expression gene analysis. (**A**) Tissue expression profiling of three proteins that were not in VGT. a, tissue expression profiling of *H2B630*. *H2B630*, histone H2B Hheb056630.1. b, tissue expression profiling of *H2B680*. *H2B680*, histone H2B Hheb056680.1. c, tissue expression profiling of *H2B740*. *H2B740*, histone H2B Hheb056740.1. FPKM, fragments per kilobase of transcript per million reads mapped; qPCR, quantitative real-time polymerase chain reaction. The R-values were calculated to analyze the correlation between the FPKM of four tissues’ transcriptomics and the relative expression levels of qPCR, in order to describe their trends of variation in four tissue samples. The *p*-value represented the probability value. OV, ovaries; VG, venom glands; DG, Dufour’s glands; CA, carcasses; VGT, venom glands’ transcriptome. (**B**) The entire transcriptome expression profiles of 200 proteins found in the venom glands transcriptome. The green dots are down-regulated proteins (Down, Log_2_ (FPKM_VG/FPKM_CA) ≤ −1, *p* < 0.05), the red dots are up-regulated proteins (UP, Log_2_ (FPKM_VG/FPKM_CA) ≥ 1, *p* < 0.05), and the blue dots are proteins that do not change significantly (NotSig). The deepening points are the proteins from ACV. ACV, artificially collected venom from mimicked hosts; FPKM, fragments per kilobase of transcript per million reads mapped.

**Figure 5 toxins-15-00377-f005:**
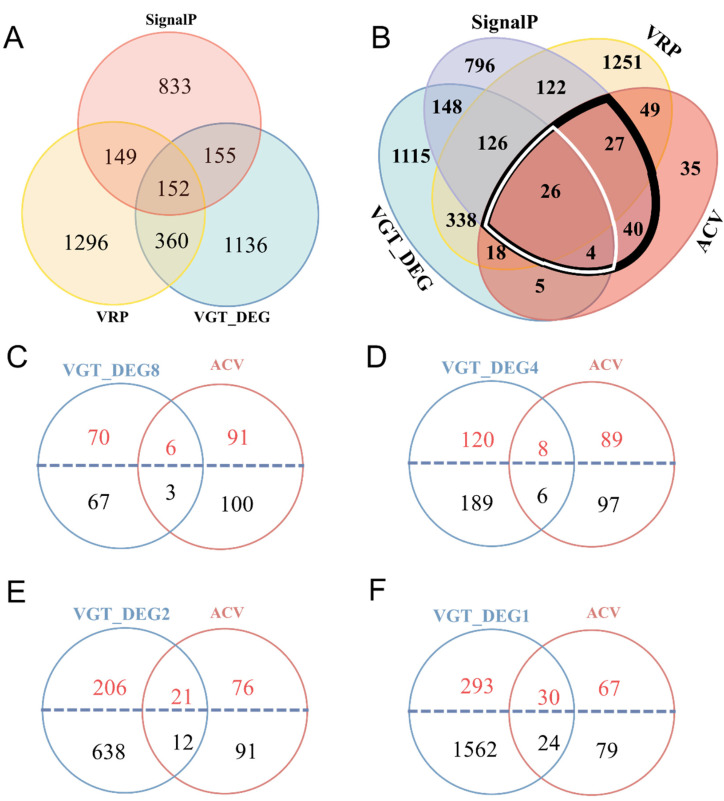
Putative venom proteins identified using combined analyses of the venom gland transcriptome and venom reservoir proteome compared to the artificially collected venom (ACV) proteome. (**A**) Venn diagram of putative venom proteins using combined transcriptomic and proteomic analyses. VG, venom gland; CA, carcasses; VRP, venom reservoirs proteome; SignalP, putative proteins with signal peptides found in the venom glands transcriptome and VRP; VGT_DEG, the proteins encoded by differentially expressed genes (DEGs) in the VG vs. CA transcriptome (log_2_(FPKM_VG/FPKM_CA) ≥ 1, *p* < 0.001). (**B**) Venn diagram of putative venom proteins using combined analyses of VG_DEG, SignalP, VRP, and ACV proteome. (**C**–**F**) Venn diagram of putative venom proteins using combined analyses of VGT_DEG and ACV proteome, as VGT_DEG = log_2_ (FPKM_VG/FPKM_CA) ≥ 8, ≥4, ≥2, ≥1, respectively, at *p* < 0.05.

**Figure 6 toxins-15-00377-f006:**
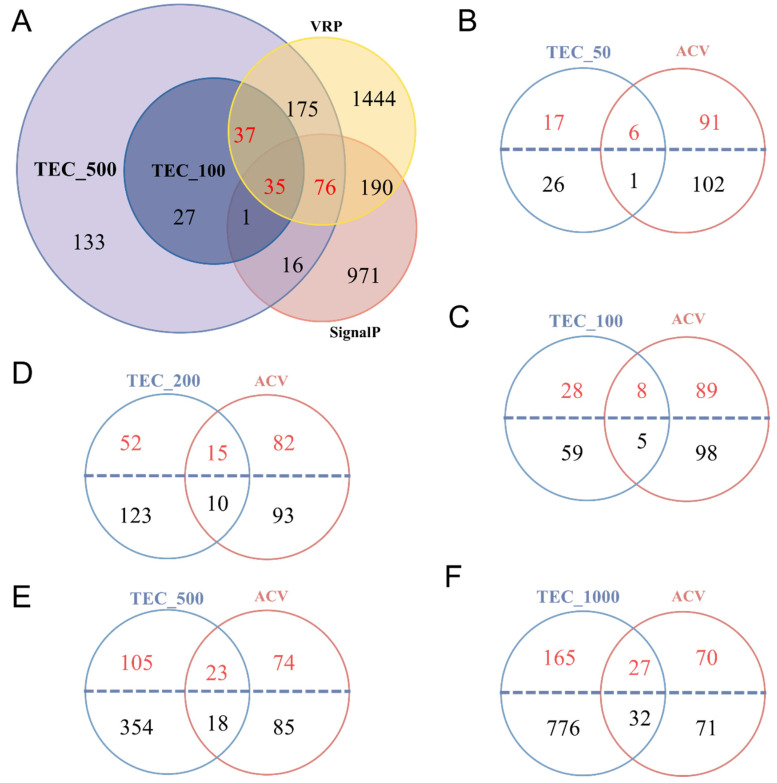
Putative venom proteins identified using Martinson’s venom protein screening criteria compared to artificially collected venom (ACV) proteome. (**A**) Venn diagram of putative venom proteins using combined analyses of transcriptomic and proteomic data and signal peptide prediction. VRP, venom reservoirs proteome; SignalP, putative proteins with signal peptides found in the venom gland transcriptome; TEC_100, the top 100 contigs (or unigenes) expressed in the venom gland transcriptome; TEC_500, the top 500 contigs expressed in the venom gland transcriptome. Red Arabic numerals indicates the number of putative venom proteins. (**B**–**F**) Venn diagram of putative venom proteins using combined analyses of the top contigs expression (TEC) information and ACV proteome. TEC_50, TEC_100, TEC_200, TEC_500, and TEC_1000: the top 50, 100, 200, 500, and 1000 contigs expressed in the venom gland transcriptome, respectively. The proteins above the blue dotted line were predicted to be with signal peptides, and pink Arabic numerals indicate the number of proteins with signal peptides.

**Figure 7 toxins-15-00377-f007:**
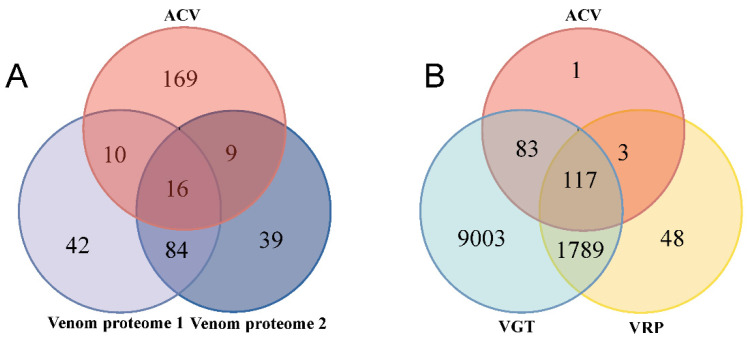
Comparison of putative venom proteins identified using different methods. (**A**) Venn diagram of putative venom proteins identified using three different methods. Venom proteome 1: 152 putative venom proteins were identified using Method 2 as mentioned in Figure 5A; Venom proteome 2: 148 putative venom proteins were identified using Martinson’s screening criteria in Method 3 described in Figure 6A. ACV, the artificially collected venom proteome collected from mimicked hosts. (**B**) Venn diagram of putative venom proteins identified using a combination of transcriptomic and proteomic analyse, the venom gland transcriptome; VRP, the venom reservoirs proteome; ACV, the artificially collected venom proteome from mimicked hosts.

**Figure 8 toxins-15-00377-f008:**
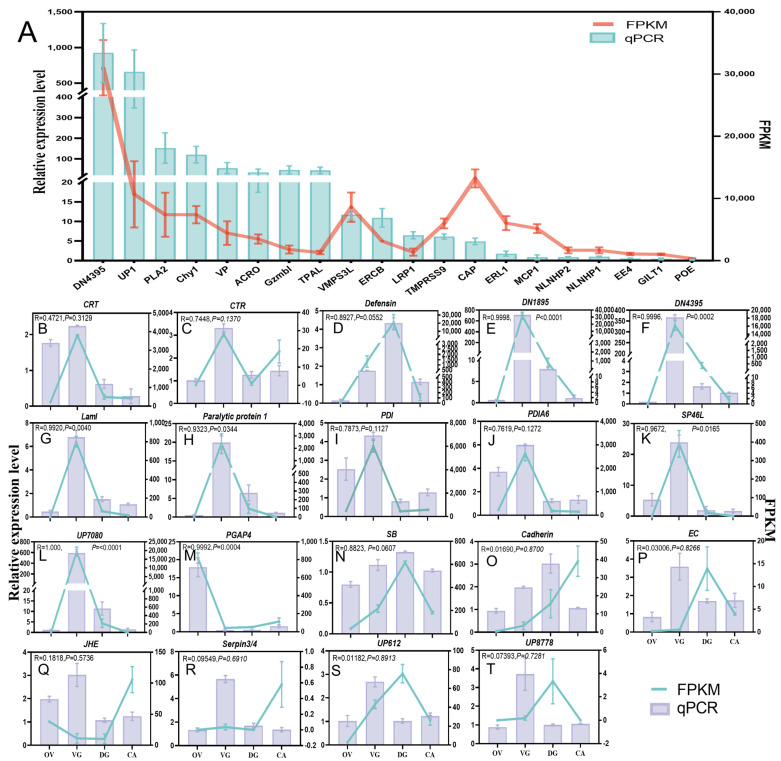
The 39 selected putative venom proteins were validated by qPCR. (**A**) The expression levels of the 20 putative venom proteins in VG were normalized to their mean expression levels in CA and are shown as mean ± standard deviations. OV, ovaries; VG, venom glands; DG, Dufour’s glands; CA, carcasses; FPKM, fragments per kilobase of transcript per million reads mapped; qPCR, quantitative real-time polymerase chain reaction. *DN4395*, putative uncharacterized protein DN4395; *UP1*, unknown protein 1; *PLA2*, phospholipase A2; *Chy1*, Chymotrypsin-1; *VP*, venom protein; *ACRO*, acrosin protein; *Gzmbl*, granzyme B-like; *TPAL*, trypsin alpha-like; *VMPS3L*, venom metalloproteinase 3-like; *ERCB*, endoplasmic reticulum chaperone BiP; *LRP1*, leucine-rich repeat-containing protein 15-like 1; *TMPRSS9*, transmembrane protease serine 9-like; *CAP*, putative capsid; *ERL1*, endoplasmic reticulum lectin 1 isoform X1; *MCP1*, mast cell protease 1A-like; *NLNHP2*, neurogenic locus notch homolog protein 2; *NLNHP1*, neurogenic locus notch homolog protein 1; *EE4*, esterase E4-like; *GILT1*, gamma-interferon-inducible lysosomal thiol like protein 1; *POE*, protein obstructor-E. These 20 genes were putative venom proteins predicted by Methods 2 and 3 in the Materials and Methods section. (**B**–**T**) The expression levels of the 19 putative venom proteins in VG, DG, and OV were normalized to their mean expression levels in CA and are shown as mean ± standard deviations. *CRT*, calreticulin; *CTR*, chymotrypsin-2; *DN1895*, putative uncharacterized protein DN1895; *LamI*, lysosomal alpha-mannosidase isoform X1; *PDI*, protein disulfide-isomerase; *PDIA6*, protein disulfide-isomerase A6 homolog; *SP46L*, serine protease 46-like; *UP7080*, uncharacterized protein Hheb070800.1; *PGAP4*, post-GPI attachment to proteins factor 4; *SB*, serine proteinase stubble. *Cadherin*, cadherin-like protein; *EC*, endochitinase; *JHE*, juvenile hormone esterase; *Serpin3/4*, serine protease inhibitor 3/4; *UP612*, uncharacterized protein Hheb006120.1; *UP8778*, uncharacterized protein Hheb087780.1. Primers are listed in Appendix A. These 19 genes were randomly selected from a set of putative venom proteins identified by Methods 1, 2, and 3 in the Materials and Methods section. The R-values were calculated to analyze the correlation between the FPKM of four tissues’ transcriptomics and the relative expression levels of qPCR, in order to describe their trends of variation in four tissue samples. The *p*-value represents the probability value.

## Data Availability

All data sets generated for this study are included in the article/Appendix A and the National Genomics Data Center with accession number PRJNA971361 (https://dataview.ncbi.nlm.nih.gov/object/PRJNA971361?reviewer=cioea8uq3a0t76hm3tonvplcgo, accessed on 14 May 2023).

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
