# Peer review of "Multi-Omic Identification of Venom Proteins Collected from Artificial Hosts of a Parasitoid Wasp"

_toxins, 2023, doi:10.3390/toxins15060377_

Round 1

Reviewer 1 Report

The paper "Multi-omic Identification of Venom Proteins Collected from 2 Artificial Hosts of a Parasitoid Wasp " gives a detailed account of the experimental work carried out by the authors. The methodology describes three approaches to the study of the proteins present in the venom, comparing all of them they have seen that they are complementary and that none of them is able to identify all the proteins.

The discussion reflects the results obtained and is supported by previous analyses. As the authors point out, this study opens the possibility of further investigation of the proteome and its mechanism of action.

Author Response

Dear Reviewer,

On behalf of all authors, I am pleased to submit a revision of our manuscript, entitled “Multi-omic Identification of Venom Proteins Collected from Artificial Hosts of a Parasitoid Wasp” (Manuscript ID: toxins-2425288). Thank you for reviewing the manuscript and for providing comments and suggestions. We took all comments and suggestions seriously in a revision. Detailed point-by-point responses to the reviewers’ comments was listed. We had highlighted the revised portions in the manuscript with a gray background.

Comments and Suggestions of Reviewer 1 for Authors

The paper "Multi-omic Identification of Venom Proteins Collected from 2 Artificial Hosts of a Parasitoid Wasp " gives a detailed account of the experimental work carried out by the authors. The methodology describes three approaches to the study of the proteins present in the venom, comparing all of them they have seen that they are complementary and that none of them is able to identify all the proteins.

The discussion reflects the results obtained and is supported by previous analyses. As the authors point out, this study opens the possibility of further investigation of the proteome and its mechanism of action.

Response:  Thank you for taking the time to read our paper and providing feedback. We are glad to hear that you have acknowledged our experimental work and appreciate your evaluation of our methodology. Our study aimed to identify venom proteins from a parasitoid wasp using three different methods, and we compared the strengths and weaknesses of all three methods. We value this result, and have discussed it in detail in the Discussion section. Additionally, we recognize that this study opened up the possibility for further exploration of the venom proteome and its mechanisms, and we will continue to conduct more in-depth investigations. Once again, thank you for your feedback and valuable suggestions. We will carefully consider your suggestions and strive to improve our research in future studies.

Yours sincerely,

Reviewer 2 Report

Reference:  “Multi-omic Identification of Venom Proteins Collected from Artificial Hosts of a Parasitoid Wasp” Submitted to TOXINS, May, 2023.

General comments: In the presented manuscript, the authors are studying the proteins of the venom of the parasitoid wasp Habrobracon hebetor, responsible for infestations of insect larvae, and a potential biological agent in the control of agricultural pests. Throughout the work, the authors describe a new method of venom collection, where an artificial host was used, which consists of an amino acid solution encapsulated in a paraffin membrane, where the studied wasps inject their venom. The authors performed mass spectrometry analyzes of the material collected by this method. Analyzes of the transcriptomes of samples obtained from the venom producing glands and other tissues of the wasps were also carried out. Throughout the text, the authors describe the identification of proteins found in the samples. In my opinion, the text brings great news, which is a new method for collecting wasp venom, but which could also be used to collect venom from other venomous animals. However, in my opinion the authors should revise the text considering that many of the proteins described are not in fact toxins, but have a great chance of being cellular contaminants obtained together with the toxins. The subject falls within the scope of TOXINS, the text is well written and the data are relevant. Attached, I send some suggestions that may make a revised text more attractive and complete.

 Specific Comments:

1- In the line 20, Key words I would introduce:  Habrobracon hebetor venom proteins…..

2- Between lines 21 and 25, Key Contributions, in addition to what has been written, which is emphasizing comparative analyzes between the venoms collected by the new method and tissue samples from the venom-producing wasp, I suggest that the authors also comment on other advantages of the method, such as yield of the collected venom, ease of collection and safety of the collector, in relation to conventional methods such as extraction by electric current applied to animals, or extraction of stingers with death of animals.  

3- In the lines 72 and 73 the authors wrote …. However, venom proteins, identified by the methods mentioned above, may not be entirely genuine components injected into the host during parasitism. Here the authors could cite some types of false toxins to exemplify the text.  

4- In th line 90 the authors wrote …. arginine kinase proteins were isolated (38). The word isolated conveys the idea of ​​purified molecules, and I don't know if this was the case. Maybe replace with identified, or described, if not the case of purified.  

5- I liked the text written in the introduction to this manuscript. Very comprehensive, clear and elucidative from a scientific point of view.  

6- I suggest authors split figure 1 into at least two or three new figures. The figures are too close together, which makes it difficult to interpret the data shown, causing loss of information.

7- Since in the new figure 1, the authors could show the morphological aspects of the H. hebetor wasp studied, interacting with a parasitized larva, in addition to the similarity with the new suggested venom collection model (this could even be in the introduction of the text to facilitate the readers' understanding of the group's goals).  

8- In the new figure 2, the authors could show the electrophoresis of the collected protein samples. The captions of course should be changed.  

9- Also in the new figure 2, the data shown for eletrophoresis (letters C and D) should be completed in the legend, pointing The polyacrylamide concentration,  if the sample was reduced as conventional biochemical analysis, and replace the word Markers by Molecular mass markers or simply MMM.  

10- The best analysis in this case for SDS-PAGE would be to show a linear gradient gel in polyacrylamide concentration, ranging from 3 to 20% in polyacrylamide concentration.  

11- Any special reason to show figure C data in blue, and figure D data in black and white?  

12- The other graphs of the old figure 1 could be shown in the new figure 3. In the letter G define the meanings of R 0.7323 and P 0.1594 ?  

13- In the lines 114 and 115 the authors wrote …. And the most abundant bands were at 66.2kDa, 25kDa and 14.4kDa. These proteins were digested by trypsin and identified using LC-MS/MS. It is unclear whether only the most abundant proteins or were all proteins analyzed in mass spectrometry.  

14- The authors wrote between lines 120 to 122 … A total of 204 single putative proteins were identified from ACV proteome, of which 200, 176 and 157 putative protein encoding genes were also expressed in VG, DG and OV transcriptomes, respectively (Figure 1E). Could the authors explain and even discuss the discrepancies found?  

15- Regarding figure 1E, in my opinion the authors could detail the data described. Perhaps in a new figure. Indicating more details of the enzymes found (proteases, nucleases, phospholipases, glycosidases, among others), the same thing for the protease inhibitors (serine proteases, metalloproteases, acid proteases, among others).  

16- In the line 124, what authors mean with the word: other proteins? Some detail must be included.  

17- Could the discrepancies shown for Histone identifications in figure 1E and 2 be explained by these proteins being cellular proteins rather than true toxins. Does the venom have holocrine secretion? Releasing cells along with the toxins?  

18- In the legend of figure 2, even if the authors have done this in the text, they must write the meanings of  OV, DG, VG and CA.  

19- in the line 179 …. As described in Method 2, better to change by …as described in Materials and Methods for Method 2 ….  

20- The text written between lines 192 to 198 and shown in supplementary figure S4 in my opinion should be part of the main text, as it describes in detail the families of proteins described in the venom-producing gland.  

21- After careful reading of the text, a great concern arises. These highthroughput proteomics and transcriptomics methods are extremely sensitive in their detection of transcripts and peptides. By the sample extraction methods there are always contaminations with cellular contents. It is difficult to come to a conclusion whether the proteins that have been sequenced, or that have had their genes sequenced, are in fact toxins, or cellular contamination. This partly explains the large discrepancies between different publications and even those obtained during this work. Without functionality analysis and, I understand that this was not the objective of this work, it is difficult to determine whether the studied molecules are indeed toxins. It's something to think about. Dozens of publications look like the one described here, with data from proteomics, transcriptomics and bioinformatics studies, but almost never prove the functionalities of the studied molecules. This is the ultimate challenge. Purification of toxins, or heterologous expression and functional studies.  

22- This is the case of the data described in figure 3. I don't know if other authors were to study this same material, they would be able to reproduce the data described in figure 3. Without studies of functionalities, they are just numbers of molecules identified in certain samples, but which lack experimental confirmation for functionality and finally proves their toxins nature.  

23- Theoretically, and considering qualitative analyses, the results of VR proteomes and ACV proteomes should not be the same? After all, didn't the ACV come from the material present in VR? Where did the different proteins found in ACV come from? Are they cellular contaminations?  

24- In this sense, the authors should make a detailed discussion in the text about possibilities of contamination of samples with cellular material, the true biological meaning of this and finally discuss the limits of the methodologies used in relation to the objectives of the study and their functional interpretations.  

25- The data shown in figures 4 and 5 are different methodologies for comparative analyzes of proteins obtained from different compartments of Habrobracon hebetor, which have the venom, having the ACV as the final reference for injection into the insect larva or the artificial collection method. The discrepancies obtained, in addition to those discussed by the authors, and may also represent cellular contamination, as already mentioned. How many times has this data been repeated? How many animals were used in the studies?  

26- Many of the differences described and even discussed by the authors also refer to molecules with very low expression in the samples, and once again this can mean contamination, fragmentation, with no biological meaning.  

27- See some examples of molecules characterized as venom components and that were detected by qPCR from materials obtained from venom-producing glands (legend to figure 6): endoplasmic reticulum chaperone BiP; leucine-rich repeat-containing protein 15-like 1; transmembrane protease serine 9-like; CAP, putative capsid; ERL1, endoplasmic reticulum lectin 1 isoform X1; MCP1, mast cell protease 1A-like; neurogenic locus notch homolog protein 2; neurogenic locus notch homolog protein 1; gamma-interferon-inducible lysosomal thiol like protein 1. A very big change from being cellular components extracted together with the transcripts of toxins, but that without legitimizing their functionality, they cannot be considered components of the venom.  

28- Throughout the text, the authors could detail whether the presence of amino acids in the solution used in the artificial host has any inhibitory activity on the biological activities of venom toxins, which could make its potential biotechnological use unfeasible.  

29- It would also be interesting for the authors to comment on whether these wasps have any potential aggressive effects on humans.  

30- Some comments if this venom has potential pharmacological actions, which may indicate possible biotechnological applications for humans, such as thrombolytic, anti-inflammatory, antioxidant agents, among others.  

31- Authors should be very careful with abbreviations used throughout the text, but not defined the first time they are used. Remember that an abbreviation may seem banal to a researcher in an area, but not to a researcher in another area. A scientific article must follow rules and this is one of them. Do a general review of the text and indicate all definitions of abbreviations in their first citation throughout the text! Excessive abbreviations without definitions tire readers and reduce the attraction of reading the text.   

32- On line 520, the authors describe how the tissues were extracted. Although they indicate that around 200 VG were extracted, the numbers for the other tissues structures studied were not shown. This should be part of the text to find out if they were really significant to the point that comparative analyzes can be made.  

33- In lines 574 and 575, the authors indicate how the structures for artificial collection of venous ACV were prepared and indicate that solutions of the amino acids leucine, phenylalanine and histidine were used. Any special reason for use these specific amino acids? Why the concentrations used? Any literature to justify this?  

34- Although the authors indicate between lines 578 and 579 that proportions of 300:1 were used between wasps and artificial structures for venom collections, it is not clear how many artificial structures containing venom were actually used for analyzes. This should be described in the text.

35- Any special reason the authors used two methods for protein determinations .... Modified Bradford Protein Assay Kit, line 568 and 569 for venom proteins. And the bicinchoninic acid (BCA) protein kit, lines 584 and 585 to determine proteins of the artificial system. Is it because of the presence of amino acids that can mask the analyzes by the Coomassie Blue method?

Author Response

Dear Editor,

On behalf of all authors, I am pleased to submit a revision of our manuscript, entitled “Multi-omic Identification of Venom Proteins Collected from Artificial Hosts of a Parasitoid Wasp” (Manuscript ID: toxins-2425288). Thank you for reviewing the manuscript and for providing comments and suggestions. We took all comments and suggestions seriously in a revision. Detailed point-by-point responses to the reviewers’ comments was listed. We had highlighted the revised portions in the manuscript with a gray background.

Comments and Suggestions of Reviewer 2 for Authors

General comments: In the presented manuscript, the authors are studying the proteins of the venom of the parasitoid wasp Habrobracon hebetor, responsible for infestations of insect larvae, and a potential biological agent in the control of agricultural pests. Throughout the work, the authors describe a new method of venom collection, where an artificial host was used, which consists of an amino acid solution encapsulated in a paraffin membrane, where the studied wasps inject their venom. The authors performed mass spectrometry analyzes of the material collected by this method. Analyzes of the transcriptomes of samples obtained from the venom producing glands and other tissues of the wasps were also carried out. Throughout the text, the authors describe the identification of proteins found in the samples. In my opinion, the text brings great news, which is a new method for collecting wasp venom, but which could also be used to collect venom from other venomous animals. However, in my opinion the authors should revise the text considering that many of the proteins described are not in fact toxins, but have a great chance of being cellular contaminants obtained together with the toxins. The subject falls within the scope of TOXINS, the text is well written and the data are relevant. Attached, I send some suggestions that may make a revised text more attractive and complete.

Specific Comments:

1-In the line 20, Key words I would introduce:  Habrobracon hebetor venom proteins…..

Response: Thank you for your suggestion. We have added “Habrobracon hebetor venom proteins” as one of the keywords in line 20 of our manuscript. We appreciate your input and hope that this addition will help to better characterize our study.

2- Between lines 21 and 25, Key Contributions, in addition to what has been written, which is emphasizing comparative analyzes between the venoms collected by the new method and tissue samples from the venom-producing wasp, I suggest that the authors also comment on other advantages of the method, such as yield of the collected venom, ease of collection and safety of the collector, in relation to conventional methods such as extraction by electric current applied to animals, or extraction of stingers with death of animals.

Response: Thank you for your suggestion. Relevant content has been added to the key contributions in line 25-27.

3- In the lines 72 and 73 the authors wrote …. However, venom proteins, identified by the methods mentioned above, may not be entirely genuine components injected into the host during parasitism. Here the authors could cite some types of false toxins to exemplify the text.

Response: Thank you for your suggestion. I had given an example to exemplify the text in line 76-78.

4- In th line 90 the authors wrote …. arginine kinase proteins were isolated (38). The word isolated conveys the idea of ​​purified molecules, and I don't know if this was the case. Maybe replace with identified, or described, if not the case of purified.  

Response: While the researchers in 2016 obtained the full-length ORF of highly expressed genes such as arginine kinase through transcriptomics, but did not perform protein purification. Therefore, I had changed “isolated” to “identified.”

5- I liked the text written in the introduction to this manuscript. Very comprehensive, clear and elucidative from a scientific point of view.

Response: Thank you very much for your affirmation.

6- I suggest authors split figure 1 into at least two or three new figures. The figures are too close together, which makes it difficult to interpret the data shown, causing loss of information.

Response: We appreciate your feedback and have carefully considered your comment regarding Figure 1. After reviewing the manuscript and our data, we believe that splitting the figure into three new figures would improve the clarity and readability of the data. Therefore, we have revised the figure and now present the data in three separate figures (Figure 1, 2 and 3) that are more spaced out and easier to interpret. We hope this change will make the information presented in our manuscript more accessible to readers.

7- Since in the new figure 1, the authors could show the morphological aspects of the H. hebetor wasp studied, interacting with a parasitized larva, in addition to the similarity with the new suggested venom collection model (this could even be in the introduction of the text to facilitate the readers' understanding of the group's goals).

Response: Thank you for your question. We appreciate your feedback and agree that the new Figure 1 provides valuable insights. We will take your suggestion into consideration and consider including these findings in the results (in line 117-123) to improve the readers’ understanding of our research goals. I hope the image I modified can meet your requirements. Thanks again for your insightful comment.

8- In the new figure 2, the authors could show the electrophoresis of the collected protein samples. The captions of course should be changed.  

Response: Thank you for your comment. We have updated the Figure 2 to include electrophoresis of the collected protein samples and revised the captions accordingly. Please kindly review the new version.

9- Also in the new figure 2, the data shown for eletrophoresis (letters C and D) should be completed in the legend, pointing The polyacrylamide concentration, if the sample was reduced as conventional biochemical analysis, and replace the word Markers by Molecular mass markers or simply MMM.

Response: Thank you for your suggestion. We have updated the legend for Figure 2, indicating that we used an 8-16% denaturing polyacrylamide gel. The protein samples were denatured at 100℃ for 15 minutes. Additionally, for clarity, we used “MMM” instead of “Markers” in the figure. Please refer to the revised Figure 2, and let us know if you have further suggestions.

10- The best analysis in this case for SDS-PAGE would be to show a linear gradient gel in polyacrylamide concentration, ranging from 3 to 20% in polyacrylamide concentration.  

Response: Thank you for your suggestion. We agree that using a linear gradient gel in polyacrylamide concentration is the best approach for achieving optimal separation in sodium dodecyl sulfate polyacrylamide gel electrophoresis (SDS-PAGE) analysis. In our experiment, this part was performed by a proteomics sequencing company. I have carefully enquired about the concentration of the gel used and they have informed us that an 8-16% gradient gel was used. We have also provided the details in the Materials and Methods section. Thank you once again for your valuable feedback.

11- Any special reason to show figure C data in blue, and figure D data in black and white?

Response: Thank you for your question. The reason for displaying data in blue in Figure C and black and white in Figure D is because the protein bands in Figure C were not clear enough, so we used blue to make them more visible. On the other hand, the protein bands in Figure D were already clear enough and did not require a color change. We deeply understand that this decision may have caused confusion and we apologize for it. We also appreciate the opportunity to explain the reasoning behind this decision.

12- The other graphs of the old figure 1 could be shown in the new figure 3. In the letter G define the meanings of R 0.7323 and P 0.1594 ?

Response: We appreciate your suggestion. When updating our manuscript, we followed your advice and created the new Figure 3 accordingly. The R-values were calculated to analyze the correlation between the FPKM of four tissues transcriptomics and the relative expression levels of qPCR, in order to describe the trends of variation of them in four tissue samples. The p-value represents the probability value. We added additional explanation in line 194-196 of the manuscript.

13- In the lines 114 and 115 the authors wrote …. And the most abundant bands were at 66.2kDa, 25kDa and 14.4kDa. These proteins were digested by trypsin and identified using LC-MS/MS. It is unclear whether only the most abundant proteins or were all proteins analyzed in mass spectrometry.

Response: It was all proteins analyzed in mass spectrometry, not only the most abundant proteins. I had modified the sentence: And the most abundant bands were at 66.2kDa, 25kDa and 14.4kDa. All the proteins were digested by trypsin and identified using LC-MS/MS via the protein database from the H. hebetor genome.

14- The authors wrote between lines 120 to 122 … A total of 204 single putative proteins were identified from ACV proteome, of which 200, 176 and 157 putative protein encoding genes were also expressed in VG, DG and OV transcriptomes, respectively (Figure 1E). Could the authors explain and even discuss the discrepancies found?

Response: Thank you for your question. We believed that one potential reason for this phenomenon was that from the reproductive system structure of Habrobracon hebetor shown in Figure 1A. It can be seen that the venom glands, ovaries and Dufour’s gland are all connected to the ovipositor. Previous studies had indicated that the venom gland tissues were associated with the female reproductive system (1), and the ACV (artificial collection venom from mimicked hosts) we collected were proteins secreted into the host by the parasitoid wasps through the ovipositor. Based on our collection process, ACV proteins may be secreted from these three tissues, and the three tissue fluids may express similar protein components. Additionally, some proteins were secreted as venom in the venom gland, but are expressed in other tissues. For example, as reported by Yang et al. (2), venom serine proteases are also expressed in other tissues. Furthermore, this phenomenon could also be caused by cell contamination due to the venom secretion mechanism you mentioned, although this requires further experimental validation. We will consider this possibility in our future studies.

15- Regarding figure 1E, in my opinion the authors could detail the data described. Perhaps in a new figure. Indicating more details of the enzymes found (proteases, nucleases, phospholipases, glycosidases, among others), the same thing for the protease inhibitors (serine proteases, metalloproteases, acid proteases, among others).

Response: Thank you for your suggestion. We appreciate your interest in our research. To address your concerns, we will generate a new figure that will include more detailed information on the enzymes found in the ACV proteome, including, oxidoreductase, peptidase, phospholipases, glycosidases, among others. Additionally, we will provide more specific information on the protease inhibitors identified, including ten sepin-type and one trypsin inhibitor. The more details about the data presented in Table S3. We hope that this new figure will address your concerns and provide a more comprehensive overview of the proteomic data we have generated for this study.

16- In the line 124, what authors mean with the word: other proteins? Some detail must be included.

Response: Thank you for your question. In line 124, we used the term “other proteins” to refer to certain proteins in ACV that match functionally related information when compared with the non-redundant protein sequence database, but do not belong to other categories such as enzymes and protease inhibitors. These “Other proteins” include immune-related proteins, calmodulin; structure-related protein, actins, transport-related proteins, transporters, and storage-related proteins, arylphorin; and so on. We will add this information in the revised text to better clarify our intended meaning.

17- Could the discrepancies shown for Histone identifications in figure 1E and 2 be explained by these proteins being cellular proteins rather than true toxins. Does the venom have holocrine secretion? Releasing cells along with the toxins?

Response: Thank you for your comments and questions. We appreciate your interest in our study. The histone proteins identified in both Figure 1E and 2 were detected in ACV which collected directly from the artificial host intracellular solution. Regarding the secretion of venom, it is known that some venomous organisms, including certain reptiles and insects, undergo holocrine secretion, which involves the release of whole cells along with their contents, including venom. However, the secretion mechanism of ACV has not been fully elucidated. We are performing further studies to investigate this and will update our findings in future publications.

18- In the legend of figure 2, even if the authors have done this in the text, they must write the meanings of  OV, DG, VG and CA.  

Response: Thank you for your suggestion. The legend has been supplemented.

19- in the line 179 …. As described in Method 2, better to change by …as described in Materials and Methods for Method 2 ….  

Response: Thank you for your suggestion. It had been modified.

20- The text written between lines 192 to 198 and shown in supplementary figure S4 in my opinion should be part of the main text, as it describes in detail the families of proteins described in the venom-producing gland.  

Response: Thank you for your valuable suggestion. We also agree that the information provided in the text between lines 192 to 198 and supplementary Table S4 regarding the families of proteins described in the venom glands was important and should be included in the main text. However, we have concerns about the size of supplementary Table S4 and its potential impact on the overall readability of the manuscript. In addition, our study mainly focuses on the ACV proteins, as described in supplementary Table S3, while Table S4 mainly lists the candidate venom proteins identified using Methods 2 and 3 described in the Materials and Methods section, which only contains a small number of ACV proteins shared by these two groups of candidate venom proteins. Therefore, we decided to list it as a supplementary table to avoid affecting the length of the main text. We appreciate your feedback and hope that our explanation can clarify our decision.

21- After careful reading of the text, a great concern arises. These high throughput proteomics and transcriptomics methods are extremely sensitive in their detection of transcripts and peptides. By the sample extraction methods there are always contaminations with cellular contents. It is difficult to come to a conclusion whether the proteins that have been sequenced, or that have had their genes sequenced, are in fact toxins, or cellular contamination. This partly explains the large discrepancies between different publications and even those obtained during this work. Without functionality analysis and, I understand that this was not the objective of this work, it is difficult to determine whether the studied molecules are indeed toxins. It's something to think about. Dozens of publications look like the one described here, with data from proteomics, transcriptomics and bioinformatics studies, but almost never prove the functionalities of the studied molecules. This is the ultimate challenge. Purification of toxins, or heterologous expression and functional studies.  

Response: Thank you very much for your careful reading of our article and for raising this issue. We shared your concern. As we pointed out in the Discussion section of our paper, it remains to be further confirmed whether the identified proteins are actually venom components. Our study aimed to identify which proteins were present in the venom glands and explored their transcription levels. From this perspective, we did not conduct in-depth studies on the potential toxicity and functionality. As you have raised, it indeed requires further experiments such as purification, heterologous expression, and functional analysis to more effectively confirm the true toxicity of the proteins. We are very grateful for your suggestion, which has sparked our contemplation and exploration.

22- This is the case of the data described in figure 3. I don't know if other authors were to study this same material, they would be able to reproduce the data described in figure 3. Without studies of functionalities, they are just numbers of molecules identified in certain samples, but which lack experimental confirmation for functionality and finally proves their toxins nature.

Response: Thank you for raising this issue, which we take very seriously. We fully understand your concern and agree that further experiments are needed to confirm the nature of the proteins identified in our study. We have described the samples and experimental procedures used to generate the data in Figure 3 in detail in the Methods section to ensure reproducibility of the results. Our study aims to identify proteins present in venom glands and explore their transcription levels. However, we acknowledge that a limitation of our study is that we did not conduct in-depth functional analysis of these proteins. We also agree that only further functional analysis can confirm the identity of these proteins as venom toxins. We appreciate your valuable suggestion, which will help us improve our future studies and increase our understanding of the toxicity of these proteins by conducting more in-depth functional studies.

23- Theoretically, and considering qualitative analyses, the results of VR proteomes and ACV proteomes should not be the same? After all, didn't the ACV come from the material present in VR? Where did the different proteins found in ACV come from? Are they cellular contaminations?

Response: Thank you for your suggestions and questions. We agree with your viewpoint. In this study, 120 ACV proteins were included in the venom reservoirs proteomes, while 84 were not. Regarding this point, we have the following speculation: Firstly, from the anatomical dissection in Figure 1, we can see that the Dufour’s gland, ovaries, and other tissues are connected with the ovipositor, and we cannot rule out the possibility that some proteins secreted by the Dufour’s gland and ovaries are present in the ACV group. In addition, the abundance of venom reservoirs proteins was much higher than that of ACV proteins. In complex sample proteomic analysis, ion suppression can seriously interfere with the ionization and detection of analytes, and therefore, some proteins may be missed in the venom reservoirs proteomes. Of course, we have also considered contamination from cell sources, as you mentioned. We speculate that protein contamination may be caused by structures similar to venom vesicles found in Drosophila Leptopilina parasitoid wasps (3). However, we did not investigate this in our study and therefore did not mention it.

24- In this sense, the authors should make a detailed discussion in the text about possibilities of contamination of samples with cellular material, the true biological meaning of this and finally discuss the limits of the methodologies used in relation to the objectives of the study and their functional interpretations.

Response: Thank you for your valuable feedback. We agree that it is important to discuss the possibilities of contamination of samples with cellular material and the true biological meaning of this in our study. We will revise the text to include a detailed discussion of these topics. Furthermore, we will also discuss the limitations of the methodologies used in relation to the objectives of our study and their functional interpretations. We appreciate your thoughtful comments and suggestions.

25- The data shown in figures 4 and 5 are different methodologies for comparative analyzes of proteins obtained from different compartments of Habrobracon hebetor, which have the venom, having the ACV as the final reference for injection into the insect larva or the artificial collection method. The discrepancies obtained, in addition to those discussed by the authors, and may also represent cellular contamination, as already mentioned. How many times has this data been repeated? How many animals were used in the studies?  

Response: Thank you for your question. The data presented in figures 4 and 5 were the result of multiple repetitions using different samples obtained from various animals. Specifically, in the experiments involving the two figures, we utilized a total of 800 H. hebetor specimens, separated into four groups of 200 each. Of these groups, three were subjected to dissections to obtain venom glands, Dufour’s glands, ovaries, and carcasses for transcriptome analysis undergoing three replicates. The remaining 200 parasitoid wasps were dissected to obtain venom reservoirs, which were then collected via high-speed centrifugation and filtered to remove any tissue debris to obtain approximately 40.2μg of protein for proteomic analysis. We recognize the importance of controlling for potential sources of contamination, and took several measures to minimize this possibility, such as washing the samples and using stringent purification protocols. However, we did discuss the possibility of cellular contamination in the manuscript and will further revise the text to provide a detailed discussion of this issue.

26- Many of the differences described and even discussed by the authors also refer to molecules with very low expression in the samples, and once again this can mean contamination, fragmentation, with no biological meaning.

Response: Thank you for your review. We understand your concern. We did mention some lowly expressed molecules in our paper, and we have acknowledged that these results may have been influenced to some extent. However, we believe that these molecules are still worth further investigation and exploration. Our samples have passed quality control checks, and standard filtering parameters were used during data analysis to minimize possible interference and contamination. Additionally, we plan to confirm the biological significance of all identified components through further functional studies, detection, and validation in future research. We will strengthen the explanations for these molecules to avoid any misunderstandings. Once again, we appreciate your valuable feedback.

27- See some examples of molecules characterized as venom components and that were detected by qPCR from materials obtained from venom-producing glands (legend to figure 6): endoplasmic reticulum chaperone BiP; leucine-rich repeat-containing protein 15-like 1; transmembrane protease serine 9-like; CAP, putative capsid; ERL1, endoplasmic reticulum lectin 1 isoform X1; MCP1, mast cell protease 1A-like; neurogenic locus notch homolog protein 2; neurogenic locus notch homolog protein 1; gamma-interferon-inducible lysosomal thiol like protein 1. A very big change from being cellular components extracted together with the transcripts of toxins, but that without legitimizing their functionality, they cannot be considered components of the venom.

Response: Thank you very much for your insightful comments. The proteins you mentioned were identified as putative venom candidate proteins by referencing some previous literature on venom protein identification methods (4-8), specifically methods 2 and 3 in the Materials and Methods section of our manuscript. And these proteins’ specific information is presented in Table S4. We used qPCR to detect the cellular components detected in these ACV proteins in our research, with the aim of highlighting the complexity of venom components and the potential regulatory mechanisms involved in venom production, as well as for validating the accuracy of our transcriptome data. We agree with the view that these molecules may not have direct venom activity individually and require further validation. In our current study, our focus is to characterize potential venom candidate proteins by combining venom gland transcriptome and ACV proteome. Further functional studies will be required to confirm the biological significance of all identified components. We will update the figure legend to clarify this and avoid any ambiguity. Thank you again for your valuable feedback.

28- Throughout the text, the authors could detail whether the presence of amino acids in the solution used in the artificial host has any inhibitory activity on the biological activities of venom toxins, which could make its potential biotechnological use unfeasible.

Response: Thank you for your comment. Amino acids were included in the solution used in the artificial host to attract the parasitic wasps. We did not investigate the potential inhibitory activity of amino acids on venom toxins in this study. However, we acknowledge that this could be an important factor to consider for the biotechnological use of venom toxins. In fact, we were able to paralyze the hosts 100% by re-injecting venom reservoir proteins and ACV into the hosts using the methods described in the manuscript. We will discuss this in the revised manuscript and propose it as a potential direction for future research.

We defined one venom reservoirs equivalent (VRE) as one venom reservoir per μL. Different concentrations of crude venom were prepared at 1, 0.5, 0.25, and 0.125 VRE, with protein concentrations of 0.83 μg/μL, 0.54 μg/μL, 0.23 μg/μL, and 0.10 μg/μL, respectively. Additionally, venom proteins collected by artificially parasitizing hosts (ACV, 0.12 μg/μL) and 1 μg/μL BSA protein were injected into fifth instar Indian meal moth larvae using a microinjection system, at a volume of 36.8 nL per head. Thirty larvae were injected with different concentrations of crude venom. Eventually, we found that all larvae were paralyzed except for those injected with BSA (Figure S1A). This was almost identical to the effect of natural parasitism (Figure S1B). We have placed more clearer new Figure S1 in both the response letter and the supplemental files for the revised manuscript. 

Due to the current difficulty in increasing the concentration of ACV protein that we have obtained, there has been limited research on the functionality of ACV protein. Going forward, we plan to conduct further in-depth research on this topic and update our results accordingly. We appreciate your suggestion and thank you once again. 

Figure S1 Effect of the venom proteins on hosts. (A)Effect of different equivalents of coarse venom and artificially collected venom proteins on hosts. VRE, venom reservoir equivalent; ACV, artificial collection venom from mimicked hosts; BSA, bovine serum albumin. (B) The unparasitized and parasitized hosts.

29- It would also be interesting for the authors to comment on whether these wasps have any potential aggressive effects on humans.

Response: Thank you very much for your valuable feedback. Your concern regarding the potential aggressive effects of the wasps on humans is indeed very important. After reviewing the literature, we did not find any reports of human attacks by the H. hebetor which used in our study. Additionally, we did not experience any aggression during the laboratory rearing of these small (2-3mm) parasitoid wasps, suggesting that they are relatively harmless. Nonetheless, we will consider evaluating their venom in future studies and provide detailed information on potential risks. Once again, we appreciate your input.

30- Some comments if this venom has potential pharmacological actions, which may indicate possible biotechnological applications for humans, such as thrombolytic, anti-inflammatory, antioxidant agents, among others.  

Response: 

Thank you very much for your valuable comment. We agreed that exploring the potential pharmacological actions of the identified venom components was a promising avenue for further research. In the early studies, it was found that the crude venom of the parasitoid wasp H. hebetor had anti-inflammatory and anti-tumor effects, as well as the ability to enhance immunity (9). These effects were achieved through the nuclear factor kappa B (NF-κB) and mitogen-activated protein kinase (MAPK) pathways, with a dose-dependent reduction in the production of nitric oxide (NO) and inhibition of pro-inflammatory mediators and cytokines, without any cytotoxic effects (9). We appreciate your suggestion and will consider it for future studies.

31- Authors should be very careful with abbreviations used throughout the text, but not defined the first time they are used. Remember that an abbreviation may seem banal to a researcher in an area, but not to a researcher in another area. A scientific article must follow rules and this is one of them. Do a general review of the text and indicate all definitions of abbreviations in their first citation throughout the text! Excessive abbreviations without definitions tire readers and reduce the attraction of reading the text.

Response: Thank you very much for your comment. We apologize for any inconvenience caused by the excessive use of abbreviations without proper definitions. We agree with your suggestion and will review the manuscript carefully to ensure that all abbreviations are defined properly at their first use. We understand that clarity and readability are essential for scientific communication, and we appreciate your feedback on this matter.

32- On line 520, the authors describe how the tissues were extracted. Although they indicate that around 200 VG were extracted, the numbers for the other tissues structures studied were not shown. This should be part of the text to find out if they were really significant to the point that comparative analyzes can be made.  

Response: Thank you for your question and feedback. We apologize for the lack of clarity regarding the extraction of tissues other than the venom gland. We extracted tissues from a total of 600 H. hebetor females, including the venom gland, ovaries, Dufour’s glands, and remnants. Each sample used approximately 200 tissues with three replicates each. We can assure you that the tissues we extracted were sufficient for transcriptome analysis, and we have a total of 12 transcriptome datasets available. The raw data has been uploaded to NCBI, and we observed differences among the tissues in terms of gene expression and pathway enrichment analyses. We have revised the manuscript to provide more information about the extraction of all tissues and will ensure that this information is included in future research. Thank you for bringing this to our attention.

33- In lines 574 and 575, the authors indicate how the structures for artificial collection of venous ACV were prepared and indicate that solutions of the amino acids leucine, phenylalanine and histidine were used. Any special reason for use these specific amino acids? Why the concentrations used? Any literature to justify this?

Response: We referred to a patent published by Yan et al. in 2020 (https://www.patentguru.com/cn/inventor/%E4%B8%A5%E6%99%BA%E8%B6%85) on the method for extracting parasitoid venom from an artificial host (202010034567.6) (10). According to the patent, the combination of leucine, phenylalanine and histidine is an effective method for extracting parasitoid venom. We selected the same combination of amino acids and tested them at similar concentration ranges. We appreciate your comments and have updated the manuscript to reflect these details.

34- Although the authors indicate between lines 578 and 579 that proportions of 300:1 were used between wasps and artificial structures for venom collections, it is not clear how many artificial structures containing venom were actually used for analyzes. This should be described in the text.

Response: Thank you very much for your message. We apologize for any confusion caused by the lack of clarity regarding the number of artificial hosts containing venom used in the text. We had carefully reviewed our records and found that in this study, we used approximately 3000 parasitoid wasps and alternately parasitized artificial hosts for 4 days, 4 hours a day, using a total of 40 artificial structures for venom collection. We collected 16.52 μg of venom proteins for mass spectrometry detection. We will modify the manuscript to make this information clearer in the text. Thank you for bringing this issue to our attention.

35- Any special reason the authors used two methods for protein determinations .... Modified Bradford Protein Assay Kit, line 568 and 569 for venom proteins. And the bicinchoninic acid (BCA) protein kit, lines 584 and 585 to determine proteins of the artificial system. Is it because of the presence of amino acids that can mask the analyzes by the Coomassie Blue method?

Response: Thank you for the comment. The reason we used two different methods for protein determinations is that each method has its own strengths and limitations. The Modified Bradford Protein Assay Kit is commonly used to quantify total protein. Therefore, we used it for detecting the concentration of venom reservoirs proteins. Nonetheless, for the artificial host, we used phenylalanine as one of its components, and we were concerned that the presence of aromatic amino acids might interfere with the detection of ACV concentration. Additionally, the concentration of ACV proteins was much lower than that of venom reservoirs proteins. Therefore, we opted to use the bicinchoninic acid (BCA) protein kit to detect the concentration of ACV in the artificial host in a more sensitive and specific manner. The choice of methods had nothing to do with potential interference from amino acids in the Coomassie Blue method. We hope this explanation clarifies our approach.

References

  1. Elias LG, Silva DB, Silva R, Peng YQ, Yang DR, Lopes NP, et al., "A comparative venomic fingerprinting approach reveals that galling and non-galling fig wasp species have different venom profiles," PLoS One, V. 13, No. 11. 2018, pp. 1-14.
  2. Yang L, Mei YT, Fang Q, Wang JL, Yan ZC, Song QS, et al., "Identification and characterization of serine protease inhibitors in a parasitic wasp, Pteromalus puparum," Scientific Reports, V. 7, Nov 16. 2017a, pp. 15755.
  3. Wan B, Poirie M, Gatti JL, "Parasitoid wasp venom vesicles (venosomes) enter Drosophila melanogaster lamellocytes through a flotillin/lipid raft-dependent endocytic pathway," Virulence, V. 11, No. 1, Jan 1. 2020, pp. 1512-21.
  4. de Graaf DC, Aerts M, Brunain M, Desjardins CA, Jacobs FJ, Werren JH, et al., "Insights into the venom composition of the ectoparasitoid wasp Nasonia vitripennis from bioinformatic and proteomic studies," Insect Molecular Biology, V. 19 Suppl 1, Feb. 2010, pp. 11-26.
  5. Yan Z, Fang Q, Wang L, Liu J, Zhu Y, Wang F, et al., "Insights into the venom composition and evolution of an endoparasitoid wasp by combining proteomic and transcriptomic analyses," Scientific Reports, V. 6, Jan 25. 2016, pp. 19604-16.
  6. Teng ZW, Xiong SJ, Xu G, Gan SY, Chen X, Stanley D, et al., "Protein discovery: combined transcriptomic and proteomic analyses of venom from the endoparasitoid Cotesia chilonis (Hymenoptera: Braconidae)," Toxins (Basel), V. 9, No. 4, Apr 12. 2017, pp. 135-59.
  7. Yang L, Yang Y, Liu MM, Yan ZC, Qiu LM, Fang Q, et al., "Identification and comparative analysis of venom proteins in a pupal ectoparasitoid, Pachycrepoideus vindemmiae," Frontiers in Physiology, V. 11. 2020, pp. 9-27.
  8. Lin Z, Wang RJ, Cheng Y, Du J, Volovych O, Han LB, et al., "Insights into the venom protein components of Microplitis mediator, an endoparasitoid wasp," Insect Biochem Molec, V. 105, Feb. 2019, pp. 33-42.
  9. Saba E, Shafeeq T, Irfan M, Lee YY, Kwon HW, Seo MG, et al., "Anti-inflammatory activity of crude venom isolated from parasitoid wasp, Bracon hebetor Say," Mediat Inflamm, V. 2017, No. 6978194. 2017, pp. 1-11.
  10. Yan ZC, Ren XY, Li YX, "A method for extracting parasitic wasp venom using artificial hosts," Book A method for extracting parasitic wasp venom using artificial hosts, 202010034567.6, Editor, ed.^eds., City, 2020, pp.
